# Solvent-free selective hydrogenation of nitroaromatics to azoxy compounds over Co single atoms decorated on Nb₂O₅ nanomeshes

Zhijun Li [1] ✉, Xiaowen Lu[1,6], Cong Guo[2,6], Siqi Ji[1], Hongxue Liu[1], Chunmin Guo[1], Xue Lu[1], Chao Wang [3], Wensheng Yan [3], Bingyu Liu[4], Wei Wu[4], J. Hugh Horton[1,5], Shixuan Xin[1] & Yu Wang [2] ✉

The solvent-free selective hydrogenation of nitroaromatics to azoxy compounds is highly important, yet challenging. Herein, we report an efficient strategy to construct individually dispersed Co atoms decorated on niobium pentaoxide nanomeshes with unique geometric and electronic properties. The use of this supported Co single atom catalysts in the selective hydrogenation of nitrobenzene to azoxybenzene results in high catalytic activity and selectivity, with 99% selectivity and 99% conversion within 0.5 h. Remarkably, it delivers an exceptionally high turnover frequency of 40377 h⁻¹, which is amongst similar state-of-the-art catalysts. In addition, it demonstrates remarkable recyclability, reaction scalability, and wide substrate scope. Density functional theory calculations reveal that the catalytic activity and selectivity are significantly promoted by the unique electronic properties and strong electronic metal-support interaction in Co₁/Nb₂O₅. The absence of precious metals, toxic solvents, and reagents makes this catalyst more appealing for synthesizing azoxy compounds from nitroaromatics. Our findings suggest the great potential of this strategy to access single atom catalysts with boosted activity and selectivity, thus offering blueprints for the design of nanomaterials for organocatalysis.

Selective hydrogenation of nitrobenzene is an important reaction used to generate valued chemicals including nitrosobenzene, phenylhydroxylamine, aniline, azobenzene, and azoxybenzene[1,2]. Of these, azoxybenzene and its derivatives are a class of compounds that have many potential applications in dyes, pharmaceuticals, polymerization inhibitors, and food additives[3]. However, the complex reduction steps and low selectivity to azoxybenzene make this reaction challenging[4,5]. Therefore, the design of a highly selective catalyst with moderate catalytic reduction abilities towards azoxybenzene is appealing. Among most of the metals, palladium,

[1]National Key Laboratory of Continental Shale Oil, College of Chemistry and Chemical Engineering, Northeast Petroleum University, Daqing, PR China. [2]Jiangsu Collaborative Innovation Centre of Biomedical Functional Materials, School of Chemistry and Materials Science, Nanjing Normal University, Nanjing, PR China. [3]National Synchrotron Radiation Laboratory, University of Science and Technology of China, Hefei, PR China. [4]National Center for International Research on Catalytic Technology, School of Chemistry and Material Sciences, Heilongjiang University, Harbin, PR China. [5]Department of Chemistry, Queen's University, Kingston, Canada. [6]These authors contributed equally: Xiaowen Lu, Cong Guo. ✉e-mail: zhijun.li@queensu.ca; yu.wang@njnu.edu.cn

iridium, and rhodium are typically used as active catalysts for selective hydrogenation reactions[6–8]. However, the high cost has greatly restricted the wide use of these precious metal-based catalysts. Moreover, strong bases or expensive organic reducing agents are typically employed in the reaction[9]. Therefore, the development of sustainable, environmentally benign, and low-cost catalysts for the selective hydrogenation of nitroaromatics to azoxy compounds is highly desirable.

Selectivity and activity in catalysis are important for efficiently producing commodity chemicals, fine chemicals, and pharmaceuticals[10]. Both factors are determined by the adsorption characteristics and activation ability of catalytically active sites toward reactants, intermediates, and products. These in turn are influenced by the geometric and electronic properties of these sites[11,12]. In homogeneous catalysis, the properties of these sites may be effectively tuned by steric and electronic structures; however, fine-tuning the selectivity in heterogeneous catalysis is challenging[10,13,14]. In addition to the catalytic activity-selectivity relationship, solvent waste removal also poses a formidable challenge for green chemical synthesis and energy consumption[15]. The use of solvent-free mechanical methods (grinding or milling) for reagents mixing and activation holds many advantages, such as shortened reaction periods, mild reaction conditions, and excellent selectivity to the target products[16–18]. Therefore, there is a strong incentive to construct highly active and selective catalyst systems that do not require the use of solvent to efficiently hydrogenate nitroaromatics to the corresponding azoxy compounds using $H_2$[19,20].

Recent years have witnessed the fast development of single atom catalysts (SACs) with unique coordination environments, high atom utilization, and appealing catalytic efficacy in a number of chemical reactions[14,21,22]. Notably, SACs possess almost 100% atomic dispersion and high activity for each active metal sites[22,23]. In addition, they can elegantly bridge heterogeneous and homogeneous catalysis to endow exceptional activity, selectively, and stability[24,25]. The catalytic performance of SACs can be improved by adjusting the coordination environment and electronic properties of metal active sites, which are generally influenced by synthetic methods and support materials[26–30]. Modulating the electron coupling between metals and supports can effectively regulate the electronic structure of metal sites for improved catalytic efficacy[31,32]. Niobium pentaoxide ($Nb_2O_5$) is an important catalyst support material due to its high stability, moderate acidity, and excellent C–O and C–C bond cleavage ability[33,34]. $Nb_2O_5$-supported metal catalysts have found applications in a variety of catalytic reactions, including hydrodeoxygenation, C–O bond activation, $C_{aromatic}$–C bonds cleavage, and aldol condensation[35–37]. However, the use of $Nb_2O_5$ as a support material for decoration of isolated non-precious metal atoms in organocatalysis has been rarely reported.

Herein, we report a facile and reliable strategy to synthesize atomically dispersed Co atoms anchored on niobium pentaoxide ($Co_1/Nb_2O_5$) nanomeshes. Aberration-corrected high-angle annular dark-field scanning transmission electron microscopy (AC HAADF-STEM), X-ray absorption spectroscopy (XAFS), and X-ray photoelectron spectroscopy (XPS) confirm the Co atoms in $Co_1/Nb_2O_5$ are atomically dispersed and positively charged. This non-precious metal-based catalyst delivers exceptional catalytic efficacy on solvent-free selective hydrogenation of nitroaromatics to azoxy compounds under base-free and solvent-free conditions (Fig. 1). The as-prepared $Co_1/Nb_2O_5$ has the potential to bypass the limitations of previously reported catalysts and enable rapid and efficient access to a diverse range of azoxy compounds from functionalized nitroaromatics. Theoretical studies reveal that the unique electronic structure and strong electronic metal-support coupling effect between Co atoms with support atoms in close proximity are beneficial for the efficient activation of reactants.

**Fig. 1 | Representative examples for the synthesis of azoxy compounds.**
**a** Oxidation of anilines into aromatic azoxy compounds. **b** Reduction of nitroarenes to aromatic azoxy compounds. **c** Hydrogenation of nitroarenes to aromatic azoxy compounds with noble metal catalyst in the presence of base. **d** This work: Direct hydrogenation of nitroarenes into aromatic azoxy compounds with non-noble metal catalysts under base-free and solvent-free conditions.

## Results

### Synthesis and characterization of atomically dispersed Co catalyst

An efficient two-step strategy (Fig. 2a) incorporating incipient wetness impregnation and microwave irradiation procedures was developed to create individually dispersed cobalt atoms over niobium pentaoxide nanomeshes ($Co_1/Nb_2O_5$). Typically, $Nb_2O_5$ was prepared by reacting ammonium niobate oxalate hydrate, melamine, and ammonium chloride in ethanol, followed by a calcination step in air. Electron microscopy characterization demonstrates that the as-prepared $Nb_2O_5$ possesses a nanomesh structure (Supplementary Figs. 1 and 2). Subsequently, the as-prepared $Nb_2O_5$ was homogeneously mixed with cobalt acetate aqueous solution by an incipient wetness impregnation approach ($Co^{2+}@Nb_2O_5$). After drying by an infrared lamp, the dried powder was microwave-treated at 800 W for 10 s to obtain $Co_1/Nb_2O_5$ with cobalt loading of 0.42 wt% and cobalt dispersion of 97% (Supplementary Table 1). This suggests that the vast majority of Co species were in the form of isolated Co atoms in $Co_1/Nb_2O_5$. Similar nanomesh morphologies between $Co_1/Nb_2O_5$ and $Nb_2O_5$ are observed by electron microscopy characterizations (Fig. 2b–d and Supplementary Fig. 3). An average height of ~6 nm is observed for $Co_1/Nb_2O_5$ as measured by atomic force microscope (AFM) imaging (Fig. 2d inset). Aberration-corrected HAADF-STEM imaging as shown in Fig. 2e displays lattice fringe spacings of 0.393 and 0.315 nm which correspond to the (001) and (180) facets of $Nb_2O_5$. The corresponding ring-like selected area electron diffraction (SAED) pattern is shown in Fig. 2e inset and agrees well with the previous reports[38,39]. A magnified AC HAADF-STEM image reveals that the isolated Co atoms are distributed over the $Nb_2O_5$ surface (Fig. 2f). Note that the cobalt atom has a lower Z contrast relative to niobium atoms. Additionally, the enlarged area provides preliminary evidence for the presence of isolated Co atoms over the $Nb_2O_5$ surface (Fig. 2f inset). Energy-dispersive X-ray spectroscopy (EDS) analysis demonstrates the homogeneously distributed O, Co, and Nb over $Nb_2O_5$ surface (Fig. 2g). By contrast, a 5.12 wt% Co NPs-containing catalyst (Co NPs/$Nb_2O_5$, Co dispersion of 39%) was created with large metallic Co species (Supplementary Fig. 4 and Table 1).

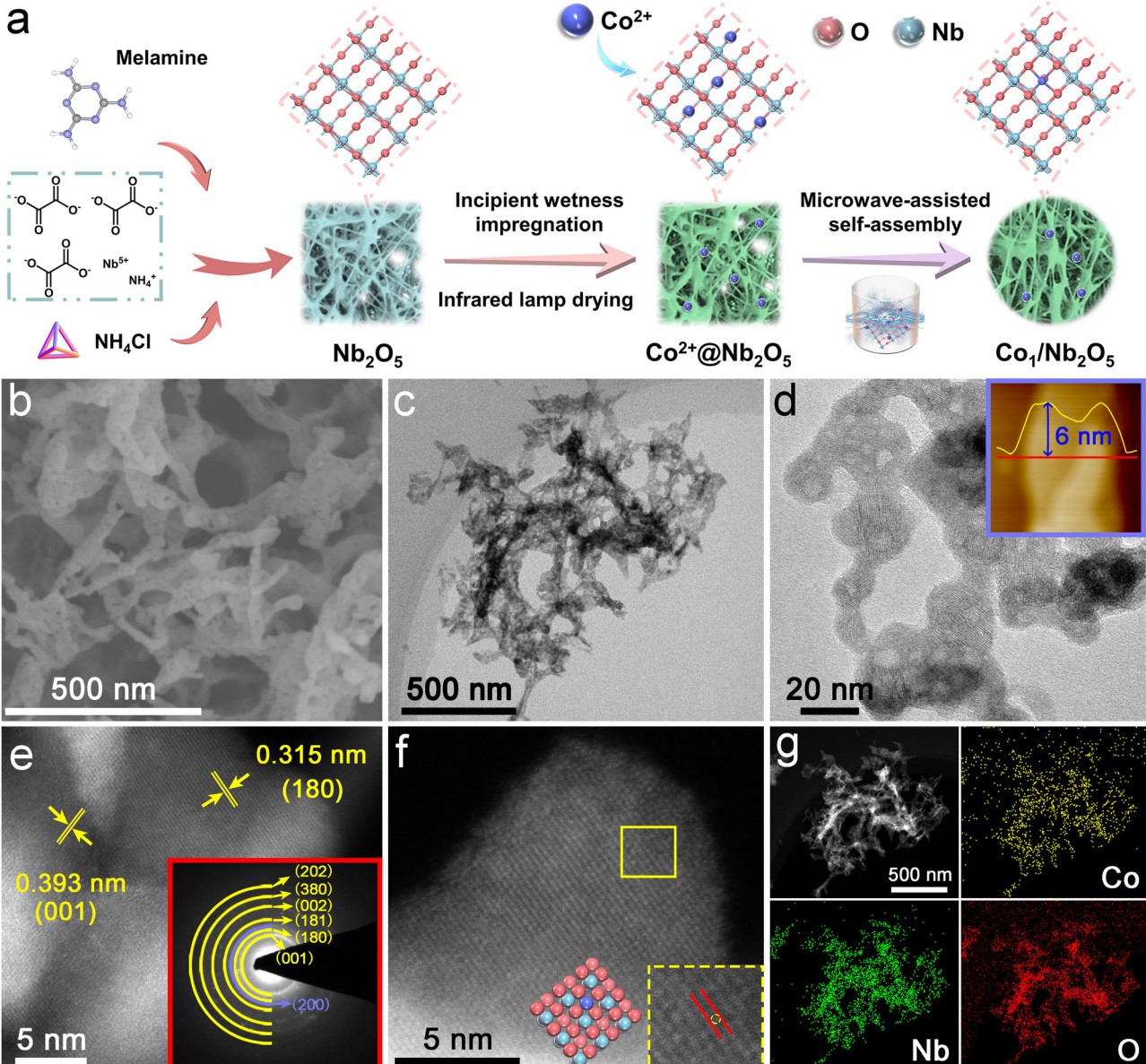

**Fig. 2 | The synthesis strategy and characterizations of Co₁/Nb₂O₅. a** Schematic illustration of the synthesis strategy. **b** SEM image. **c** TEM image. **d** HR-TEM image (the inset is an AFM image showing the height profile). **e** AC HAADF-STEM image (the inset is SAED pattern). **f** AC HAADF-STEM image at high magnification. The red, blue, and purple atoms represent O, Nb, and Co, respectively. **g** EDS mapping images.

X-ray diffraction (XRD) patterns of samples are exhibited in Fig. 3a. Typical diffraction peaks of Nb₂O₅ (JCPDS No. 30-0873) agree well with SAED patterns as shown in Fig. 2e. Moreover, there are no evident metallic cobalt peaks found in Co₁/Nb₂O₅, suggesting that these Co species are highly dispersed on the Nb₂O₅ surface, even for Co NPs/Nb₂O₅. Raman spectra are shown in Fig. 3b. The overlapped Raman signals are categorized into three band groups[40,41]: a high-wavenumber ($v_{Hi}$, 485 cm⁻¹ ~ 808 cm⁻¹), a mid-wavenumber ($v_{Mid}$, 175 ~ 370 cm⁻¹), and a low-wavenumber band group ($v_{Lo}$, 93 ~ 172 cm⁻¹). The peaks in the 100 ~ 400 cm⁻¹ region are associated with typical bending modes of Nb–O–Nb linkages[41]. After Co deposition, the intensities of peaks at 120 cm⁻¹ in Co₁/Nb₂O₅ and Co NPs/Nb₂O₅ are enhanced, while the peaks at 230 cm⁻¹ are weakened. This suggests the formation of Co–O–Nb linkages and disordering of the bending modes in Nb₂O₅[41]. In addition, the initially high-intensity peak at 690 cm⁻¹ is reduced significantly due to the Co deposition partially distorting the Nb₂O₅ structure[41,42]. X-ray photoelectron spectroscopy (XPS) was conducted to understand the electronic properties of samples (Fig. 3c

and Supplementary Fig. 5). Compared with the Co NPs/Nb₂O₅ sample (Fig. 3c), the Co 2$p_{3/2}$ peak of Co₁/Nb₂O₅ is located at 781.1 eV, agreeing well with assignment as a positively charged Co species[43,44]. The O 1$s$ and Nb 3$d$ spectra of Co₁/Nb₂O₅ and Co NPs/Nb₂O₅ are shown in Supplementary Fig. 5c. We observe surface oxygen vacancies in both samples, indicating the presence of electronic coupling interaction between the introduced Co species and the Nb₂O₅ support[45]. The Nb 3$d$ spectra of both samples can be fitted to two peaks for Nb 3$d_{5/2}$ and Nb 3$d_{3/2}$, respectively. The position of the Nb 3$d_{5/2}$ peak of Co₁/Nb₂O₅ is centered at 207.1 eV, which is slightly lower than that of Co NPs/Nb₂O₅ (207.3 eV), suggesting a relatively larger concentration of Nb⁴⁺ species and oxygen vacancies over the former[46]. Nonetheless, the chemical shift suggests that only a small portion of the niobium present in either sample is in the form of Nb⁴⁺ species. In situ CO-diffuse reflectance infrared Fourier transform spectroscopy (CO-DRIFTS) was performed to investigate atomic structures of Co species in Co₁/Nb₂O₅ and Co NPs/Nb₂O₅ (Fig. 3d). The broad bands around 2171 cm⁻¹ and 2115 cm⁻¹ are ascribed to residual gas-phase CO molecules[47,48]. For Co₁/Nb₂O₅, a

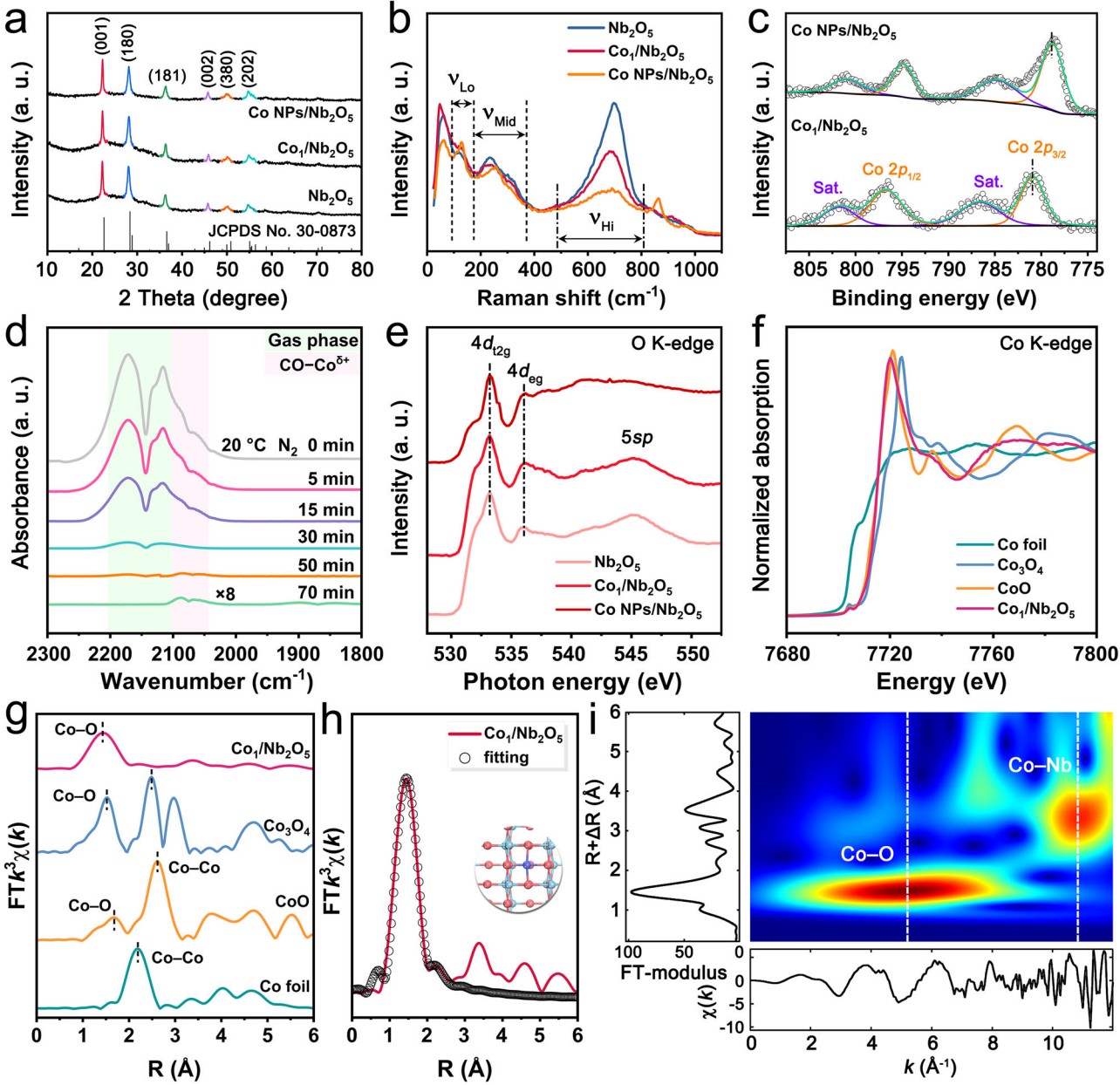

**Fig. 3 | Structural characterizations of as-prepared Co₁/Nb₂O₅. a** XRD patterns. **b** Raman spectra. **c** XPS Co 2*p* spectra. **d** In situ CO-DRIFTS of Co₁/Nb₂O₅. **e** O K-edge spectra. **f** XANES spectra at Co K-edge. **g** Fourier transformed $k^3$-weighted Co K-edge of EXAFS spectra. **h** EXAFS fitting in R space (inset is the model of Co₁–O₄). **i** 3D contour WT-EXAFS plot.

pair of CO adsorption peaks were observed at 2089 cm⁻¹ and 2067 cm⁻¹, respectively, which could be assigned to linearly adsorbed CO on positively charged Co species[49,50]. The absence of other peaks excludes the presence of Co multiatomic species in Co₁/Nb₂O₅. In the case of Co NPs/Nb₂O₅, two peaks appeared at 2015 cm⁻¹ and 1984 cm⁻¹, respectively, which are associated with the CO adsorbing on metallic Co in linear and bridge form (Supplementary Fig. 6)[51]. Electron paramagnetic resonance (EPR) spectra (Supplementary Fig. 7) demonstrate a sharp signal at a g value of 2.003 for Co₁/Nb₂O₅, which is associated with the coordinatively unsaturated Co species in the catalyst[45]. H₂-temperature-programmed reduction (H₂-TPR) results are shown in Supplementary Fig. 8. Two peaks at 508 °C and 588 °C are observed for Nb₂O₅, corresponding to surface Nb–O–Nb and interior Nb–O–Nb sites[46]. After introducing Co species, only two peaks at 341 °C and 484 °C are observed, which are associated with the reduction of Co–O–Nb and interior Nb–O–Nb species, respectively. This suggests

Nb₂O₅ can serve as a heterogeneous support for anchoring the isolated Co atoms and restricting them from agglomeration.

To understand the chemical state of Co species in the catalysts, Co L₂,₃-edge near-edge X-ray absorption fine structure (NEXAFS) results were collected with synchrotron-based soft X-ray radiation. As shown in Supplementary Fig. 9, the peak of the Co L-edge spectrum in Co₁/ Nb₂O₅ displays a positive shift of 0.9 eV relative to that of Co NPs/ Nb₂O₅, indicating a higher valence state of Co in Co₁/Nb₂O₅[52,53]. This is in good agreement with the XPS results as discussed in Fig. 3c. O K-edge NEXAFS spectra (Fig. 3e) exhibit a lowered $4d_{t2g}$ peak intensity for O–Nb coupling after deposition of Co species in Nb₂O₅. These results, in conjunction with the XPS O 1*s* spectra, demonstrate that the Co₁/Nb₂O₅ and Co NPs/Nb₂O₅ interfaces facilitate the formation of oxygen vacancies in these catalysts[54]. Synchrotron radiation-based ultraviolet photoemission spectroscopy (UPS) was performed to study the electronic state of Co₁/Nb₂O₅ and Co NPs/Nb₂O₅. As displayed in

Supplementary Fig. 10, the values of valence band maxima for $Co_1$/$Nb_2O_5$ and Co NPs/$Nb_2O_5$ are determined to be 2.27 eV and 2.56 eV, respectively. This indicates a change in the electron arrangement of Co 3$d$ orbitals, which is associated with the metal-support interaction and the coordination environment. Because the valence electrons near the Fermi level contribute greatly to the $d$ states, the change in the valence band signifies the movement of the $d$ band center. Fourier-transform infrared (FT-IR) spectra of samples are shown in Supplementary Fig. 11 and signals typical of Nb–O–Nb and Nb=O are observed. $N_2$ adsorption/desorption isotherms (Supplementary Fig. 12 and Table 1) show the specific surface area of $Nb_2O_5$, $Co_1$/$Nb_2O_5$, and Co NPs/$Nb_2O_5$ are determined to be 49.8, 56.7, and 80.0 $m^2$/g, respectively. TGA/DSC results show similar results for $Co_1$/$Nb_2O_5$ and Co NPs/$Nb_2O_5$ (Supplementary Fig. 13). This implies that the incorporation of Co species into $Nb_2O_5$ did not significantly affect the material properties.

The atomic dispersion and coordination information of Co species in $Co_1$/$Nb_2O_5$ were examined by synchrotron-radiation X-ray absorption fine structure spectroscopy (XAFS). As disclosed in Fig. 3f, the pre-edge of Co K-edge in $Co_1$/$Nb_2O_5$ is located between those of CoO and $Co_3O_4$, and closer to that of CoO. This indicates the valence state of Co species is between $Co^{2+}$ and $Co^{3+}$, though closer to $Co^{2+}$. The positively charged Co species result from the strong charge transfer between atomically dispersed Co species and the $Nb_2O_5$ support[55]. In the $k$-space of the extended X-ray absorption fine structure spectra (EXAFS), $Co_1$/$Nb_2O_5$ displays a different pattern compared with Co foil (Supplementary Fig. 14), implying they possess different coordination structures. Figure 3g shows the Fourier-transformed $k^3$-weighted EXAFS spectra of Co in $Co_1$/$Nb_2O_5$ together with other reference samples. The Co foil reference exhibits a dominant peak at 2.19 Å that is associated with Co–Co scattering in the first coordination sphere. CoO displays two peaks at 1.67 Å and 2.62 Å that can be indexed to Co–O and Co–Co scattering, respectively. $Co_3O_4$ exhibits two peaks at 1.52 Å and 2.49 Å, also corresponding to Co–O and Co–Co scattering, respectively. As for $Co_1$/$Nb_2O_5$, there is only one prominent peak centered at 1.43 Å and this is assigned to Co–O scattering in the first shell. These results confirm the atomic distribution of Co species in $Co_1$/$Nb_2O_5$. The EXAFS fitting results (Fig. 3h, Supplementary Fig. 15 and Table 2) reveal that the Co atom is encircled by oxygen atoms with an average coordination number of 4.6. Wavelet transform (WT) results (Fig. 3i and Supplementary Fig. 16) confirm the presence of Co–O bond in $Co_1$/$Nb_2O_5$, in line with the Co K-edge FT-EXAFS analysis. Together, these findings demonstrate an atomic dispersion of Co species over $Nb_2O_5$ support.

## Evaluation of catalytic performance

The catalytic performance of $Co_1$/$Nb_2O_5$ in the selective hydrogenation of nitrobenzene to azoxybenzene was initially evaluated under 1 atm of $H_2$ at 20 °C with addition of solvent (Supplementary Table 3). A series of solvents were screened and the use of tetrahydrofuran/$H_2O$ (4:1, v:v) mixed solvents secured the optimum reaction conditions (Supplementary Table 4). We observe low catalytic activity in the case of $Nb_2O_5$, implying the Co species are essential for the catalytic performance (Supplementary Table 5). With $Co_1$/$Nb_2O_5$, the reaction proceeds smoothly to give the desired azoxybenzene with high conversion (99%) and selectivity (99%) within 1.5 h (Supplementary Fig. 17 and Table 5). No side product of aniline is detected for this reaction. Various Co salts, including $Co(NO_3)_2$, Co(Ac)$_2$, $CoCl_2$, and CoPc, are unable to efficiently catalyze this transformation, resulting in low catalytic activity (Supplementary Table 5). Accordingly, an extremely high turnover frequency (TOF) value of 11524 $h^{-1}$ is noted for $Co_1$/$Nb_2O_5$ compared with the control samples. For Co NPs/$Nb_2O_5$ (Supplementary Fig. 18 and Table 5), aniline is identified as the main product with a high selectivity of 99%, together with trace amounts of azoxybenzene and azobenzene.

Next, we sought to investigate the catalytic efficacy of $Co_1$/$Nb_2O_5$ in catalyzing this reaction under solvent-free conditions. We observed an exceptionally high catalytic activity of $Co_1$/$Nb_2O_5$, with maximized atomic utilization. The reaction yielded azoxybenzene with excellent conversion (99%) and exclusive selectivity (99%) within merely 0.5 h (Fig. 4a and Supplementary Fig. 19). Additionally, the 99% selectivity remains unchanged following a further 9.5 h reaction, emphasizing excellent catalyst performance. This implies that the undesired side reaction was effectively constrained in the absence of Co–Co bonds. In the case of Co NPs/$Nb_2O_5$, aniline is the main product (Fig. 4b). This might result from the aggregated Co species over $Nb_2O_5$ with multiple Co–Co bonds. Catalysts consisting of Co single atoms anchored on other oxides using the same synthetic method with O–coordination structures were also prepared, being $Co_1$/MgO and $Co_1$/$V_2O_5$, respectively (Supplementary Figs. 20 and 21, and Table 2). A nitrogen-doped carbon support with Co single atoms ($Co_1$/N-C) was fabricated to represent different coordination environments, for example, Co–N coordination (Supplementary Fig. 22 and Table 2). Remarkably, an exceptional TOF value of 40377 $h^{-1}$ for $Co_1$/$Nb_2O_5$ was determined, significantly higher than other control samples (Fig. 4c). Poor catalytic activity and selectivity of $Co_1$/MgO, $Co_1$/$V_2O_5$, and $Co_1$/N-C are observed (Supplementary Fig. 23). This might result from the unique coordination environments, electronic structures, and electronic metal-support interactions of catalytically active Co sites in $Co_1$/$Nb_2O_5$ that influence the reaction pathways and energy barriers. Under these mild conditions, $Co_1$/$Nb_2O_5$ also demonstrates superior catalytic performance compared with previously reported catalysts (Supplementary Table 6).

Kinetic studies were performed to gain more insights into the origin of the catalytic activity of $Co_1$/$Nb_2O_5$ based on initial conversion rates of nitrobenzene at different temperatures (Fig. 4d and Supplementary Fig. 24). Compared with Co NPs/$Nb_2O_5$, a lowered activation energy ($E_a$) of 39 kJ $mol^{-1}$ was determined for $Co_1$/$Nb_2O_5$ which suggests an enhanced catalytic activity. This offers evidence that the functionalization of a moderate amount of atomically dispersed Co atoms over $Nb_2O_5$ can significantly boost the catalytic activity. After 10 cycles of repetitive use, this $Co_1$/$Nb_2O_5$ catalyst exhibits admirable stability without noticeable activity degradation (Fig. 4e and Supplementary Table 7). The crystalline structure and morphology of recycled $Co_1$/$Nb_2O_5$ catalyst do not show any evident differences (Supplementary Fig. 25). Importantly, the EXAFS results of spent $Co_1$/$Nb_2O_5$ catalyst (Supplementary Fig. 26 and Table 2) demonstrate that the dispersion and coordination environment of Co atoms are unchanged after 10 cycles. In addition, Co L-edge and O K-edge NEXAFS (Supplementary Fig. 27) reveal that no substantial electronic structural changes were observed. These results imply a strong metal-support interaction in the $Co_1$/$Nb_2O_5$ catalyst with high stability.

$H_2$ dissociation ability over catalysts plays a crucial role in the hydrogenation reactions[56–58]. $H_2$-temperature-programmed desorption ($H_2$-TPD) measurements were initially performed and the results (Supplementary Fig. 28) show that Co NPs/$Nb_2O_5$ exhibits a higher intensity of desorption peaks over $Co_1$/$Nb_2O_5$ and $Nb_2O_5$, implying the existence of a higher amount of active sites and greater $H_2$ adsorption capacity (Supplementary Table 1). The desorption temperature of $Co_1$/$Nb_2O_5$ is slightly smaller than that of Co NPs/$Nb_2O_5$, but higher than that of $Nb_2O_5$. Based on Kyriakou's work[59], the $H_2$ dissociation barrier will be lower on the active sites once the corresponding binding energy of dissociated H atoms is higher. Therefore, the higher $H_2$ desorption temperature of Co NPs/$Nb_2O_5$ suggests that it favors the activation and dissociation of $H_2$ more efficiently than $Co_1$/$Nb_2O_5$ and $Nb_2O_5$. The $H_2$ dissociation activity of the samples was further evaluated using an $H_2$-$D_2$ exchange experiment (Supplementary Fig. 29). The HD formation rate follows the order Co NPs/$Nb_2O_5$ > $Co_1$/$Nb_2O_5$ > $Nb_2O_5$. Co NPs/$Nb_2O_5$ achieved a higher HD

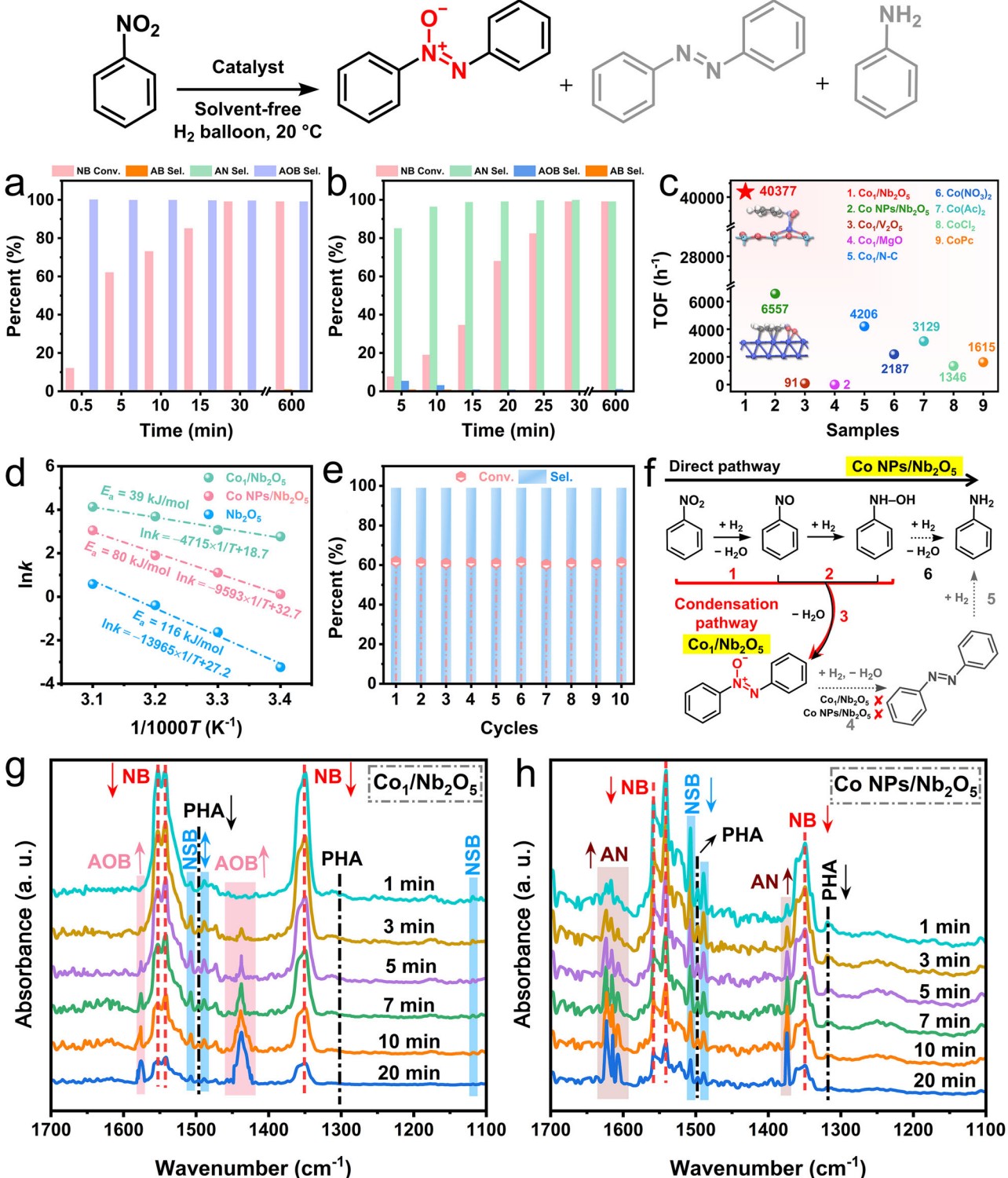

**Fig. 4 | Catalytic performance of Co₁/Nb₂O₅ in the solvent-free selective hydrogenation of nitrobenzene to azoxybenzene.** Conversion and selectivity of (**a**) Co₁/Nb₂O₅ and (**b**) Co NPs/Nb₂O₅. **c** The corresponding TOF values of different samples. **d** Arrhenius plots and *E*a values. **e** Recycling results (Reaction time: 5 min). **f** The proposed reaction routes. In situ DRIFTS spectra were recorded during the hydrogenation of nitrobenzene over (**g**) Co₁/Nb₂O₅ and (**h**) Co NPs/Nb₂O₅. (NB nitrobenzene; NSB nitrosobenzene; PHA phenylhydroxylamine; AB azobenzene; AOB azoxybenzene; AN aniline).

formation rate than those of Co₁/Nb₂O₅ and Nb₂O₅, suggesting that the addition of Co species could considerably enhance the H₂ dissociation activity and subsequently promote the hydrogenation reactions. Although Co NPs/Nb₂O₅ exhibits excellent nitrobenzene conversion, it exhibits extremely poor azoxybenzene selectivity. This implies that the difference in H₂ dissociation activity might not be the only reason

affecting the overall catalytic performance of the catalyst. More discussion on the origin of the selectivity difference can be found in the kinetics simulations of key hydrogenation steps in the Mechanism investigation section.

Based on these experimental findings, we assume that the Co₁/Nb₂O₅ follows the steps from 1-2-3, while the Co NPs/Nb₂O₅ take the

steps from 1-2-6 under the reaction conditions (Fig. 4f). To confirm this point, the selective hydrogenation of nitrobenzene was examined on $Co_1/Nb_2O_5$ and Co NPs/$Nb_2O_5$ by in situ DRIFTS. In the case of $Co_1/Nb_2O_5$ (Fig. 4g), the spectra show a slow consumption of nitrobenzene (IR bands at 1552, 1543, and 1349 cm$^{-1}$)[60] and the appearance of azoxybenzene with rapidly increased IR bands[9,61,62] at 1576 and 1437 cm$^{-1}$. The intermediate species of phenylhydroxylamine (1496 and 1302 cm$^{-1}$)[9] and nitrosobenzene (1507 and 1119 cm$^{-1}$)[9,61] were also observed. For Co NPs/$Nb_2O_5$ (Fig. 4h), the IR bands of nitrobenzene decrease gradually, while the bands of aniline (1623, 1614, 1606, and 1375 cm$^{-1}$)[9,61] increase, indicating H$_2$ introduction favors the formation of aniline. In addition, we also observed the key reaction intermediate of nitrosobenzene and phenylhydroxylamine. These IR results are in excellent agreement with the experimental results. To validate the utility of the $Co_1/Nb_2O_5$ catalyst in large-scale organocatalysis, we scaled up the reaction (50-fold) to the gram scale and established optimized conditions (Supplementary Fig. 30). The results show the catalytic performance of $Co_1/Nb_2O_5$ was nearly identical to the lab-scale to give azoxybenzene, underscoring the potential of reaction scale tolerance.

With the optimized reaction conditions established, the substrate scope of solvent-free selective hydrogenation of nitroaromatics to azoxy compounds was explored. As shown in Fig. 5 and Supplementary Figs. 31–46, a variety of nitroaromatics were tested and all of these substrates are well tolerated and converted efficiently to the corresponding azoxy compounds in high conversion (up to 99%) and selectivity (up to 99%). The nitroaromatics bearing either electron-withdrawing substituents (halogen group, and nitro group) or electron-donating substituents (hydroxyl group, methoxyl group, methyl group, and amino group), situated at the *para*-position of the aromatic ring, did not have a noticeable impact on the reaction output. Note that methyl and methoxy groups at either *ortho*- or *meta*-positions (2r, 2o, 2l, 2j) underwent smooth conversion to afford the desired products with slightly lower efficacy compared to the more sterically accessible ones at para-position (2a, 2b). This might be due to the steric hindrance effect of the substituent groups that unfavor the coupling reactions. In addition to symmetrical azoxy compounds, we also tried to explore the potential feasibility of synthesizing products with unsymmetrical compounds (Supplementary Fig. 47). To our delight, this $Co_1/Nb_2O_5$ catalyst shows good reactivity and affords the desired unsymmetrical compounds in moderate yields in most cases. Together, the results demonstrate the developed $Co_1/Nb_2O_5$ catalyst can efficiently catalyze solvent-free selective hydrogenation of nitroaromatics to yield azoxy compounds.

## Mechanism investigation

To explore the origin of the high catalytic performance of $Co_1/Nb_2O_5$ catalyst, density functional theory (DFT) calculations were performed. The optimized geometric structures of $Nb_2O_5$, $Co_1/Nb_2O_5$, and Co NPs/$Nb_2O_5$ are shown in Supplementary Figs. 48–50. Based on AC STEM and XAFS characterizations, we embedded a Co atom on the $Nb_2O_5$(001) surface to describe $Co_1/Nb_2O_5$ and employed Co(111) to represent Co NPs/$Nb_2O_5$ for the calculation. Bader charge and charge density difference analysis of $Co_1/Nb_2O_5$ indicate that the Co atom loses 0.83 $e$ to the neighboring O atoms in $Co_1/Nb_2O_5$ (Fig. 6a and Supplementary Fig. 51). This suggests the presence of electronic metal-support interactions and the positively charged Co atoms. In addition, there is orbital mixing between O and Co atoms based on the projected density of states (PDOS) in $Co_1/Nb_2O_5$ relative to that of $Nb_2O_5$, as shown in Fig. 6b and Supplementary Fig. 52. The $d$ band centers of Co in $Co_1/Nb_2O_5$ and Co(111) were determined to be −2.95 eV and −2.09 eV, respectively (Fig. 6b, c). Figure 6d, e shows the reaction pathways and the corresponding calculated energy profiles over $Co_1/Nb_2O_5$ and Co(111). The configurations of intermediates are displayed

in Supplementary Figs. 53 and 54. The nitrobenzene adsorption energies ($E_{PhNO2}$) for $Co_1/Nb_2O_5$ and Co(111) were both calculated to be exothermic at −1.25 eV and −2.57 eV, respectively. The larger adsorption energy of Co(111) demonstrates it has a much stronger affinity to nitrobenzene, which is in good agreement with its $d$ band center that is closer to the Femi level than $Co_1/Nb_2O_5$. With $Co_1/Nb_2O_5$, the adsorbed *PhNO$_2$ can be hydrogenated to afford a *PhNO$_2$H intermediate (−0.24 eV) and then form *PhNO (−1.61 eV). Subsequently, the *PhNO intermediate would be reduced to *PhNOH and then *PhN via two hydrogenation steps, both of which are downhill in the energy profile by −0.69 eV and −0.33 eV, respectively. Finally, *PhN is converted to Ph-NNOPh with high priority by a highly exothermic process (−2.15 eV) over *PhNH (−1.70 eV). In the case of Co(111), the conversion of *PhNO$_2$ to *PhNO$_2$H is uphill in the energy profile by 0.58 eV due to the strong adsorption of PhNO$_2$. Following the formation of *PhNO$_2$H, its further hydrogenation to *PhNO is energetically favorable (−1.86 eV), and then *PhNO undergoes two successive hydrogenation steps to generate *PhN and which is overall exothermic by −1.08 eV. By overcoming an energy barrier of merely 0.07 eV, *PhN can be easily transformed to *PhNH compared with a higher energy barrier of 1.17 eV to obtain Ph-NNOPh.

We also performed a kinetic analysis to gain more insights into the reactivity of $Co_1/Nb_2O_5$ and Co(111). It was found that the kinetic barrier ($E_b$) of the dissociation of an H$_2$ molecule on $Nb_2O_5$ is 0.91 eV (Supplementary Fig. 55), consistent with the previous report (0.88 eV)[63]. The $E_b$ of $Co_1/Nb_2O_5$ (0.84 eV) is close to that of $Nb_2O_5$ (Supplementary Fig. 56); this may be due to the fact that H$_2$ is physically adsorbed on $Nb_2O_5$ and $Co_1/Nb_2O_5$, of which the adsorption energy $E_{H2}$ is −0.17 and −0.24 eV, respectively. Specially, in both $Nb_2O_5$ and $Co_1/Nb_2O_5$, H$_2$ is preferentially located above the Nb site (Supplementary Fig. 57). The H$_2$ above the Co$_1$ site is also physisorbed while being more weakly bound (−0.12 eV). Therefore, H$_2$ dissociation and PhNO$_2$ hydrogenation may occur on the $Nb_2O_5$ and the Co$_1$ site, respectively. Moreover, as expected, Co(111) exhibits stronger H$_2$ adsorption than $Co_1/Nb_2O_5$ and $Nb_2O_5$ (Supplementary Fig. 57), and its $E_b$ is as low as 0.02 eV (Supplementary Fig. 58), which is in agreement with previous studies (0.03 eV)[64]. In the presence of *PhNO$_2$, H$_2$ dissociation on Co(111) is still kinetically favorable (Supplementary Fig. 59). These modeling results are consistent with our H$_2$-TPD measurements and H$_2$-D$_2$ exchange experiments. Note that the small $E_b$ of H$_2$ dissociation on Co(111) should not affect the PhNO$_2$ adsorption and the subsequent hydrogenation steps, given the stronger adsorption of PhNO$_2$ compared to H$_2$. As H$_2$ is physisorbed, we also calculated H$_2$ dissociation on *PhNO$_2$ of $Co_1/Nb_2O_5$ via the Eley−Rideal (ER) mechanism and found a small $E_b$ of 0.34 eV (Supplementary Fig. 60).

We then moved our attention to the subsequent hydrogenation steps. For the first four hydrogenation steps, the largest $E_b$ of $Co_1/Nb_2O_5$ occurs in the conversion of *PhNOH to *PhN (0.84 eV), and that of Co(111) is the conversion of *PhNO to *PhNOH (0.95 eV). Specifically, $Co_1/Nb_2O_5$ exhibits a favorable $E_b$ of 0.22 eV for facilitating the further coupling of *PhN and *PhNO to *Ph-NNOPh, which is lower than that of the competing *Ph-NH formation (0.65 eV). By contrast, the hydrogenation of *PhN to *PhNH on Co(111) requires overcoming a moderate $E_b$ of 1.02 eV, whereas the barrier for the *Ph-NNOPh generation is 1.21 eV. It should be noted that the active sites of the niobium oxide component are essential for the dehydration reaction because the Nb site of $Co_1/Nb_2O_5$ exhibits favorable binding to*PhNO, which promotes the coupling of *PhNO and *PhN. In addition, we explored the influence of solvent on the reaction processes and found that the solvent effect did not change the product selectivity of $Co_1/Nb_2O_5$ and Co(111) (Supplementary Figs. 61 and 62). For example, after considering the solvent effect, the $E_b$ of Ph-NNOPh formation on $Co_1/Nb_2O_5$ (0.49 eV) is still lower than that of the competing *Ph-NH formation (0.66 eV). These results are consistent with the experimental observations. The

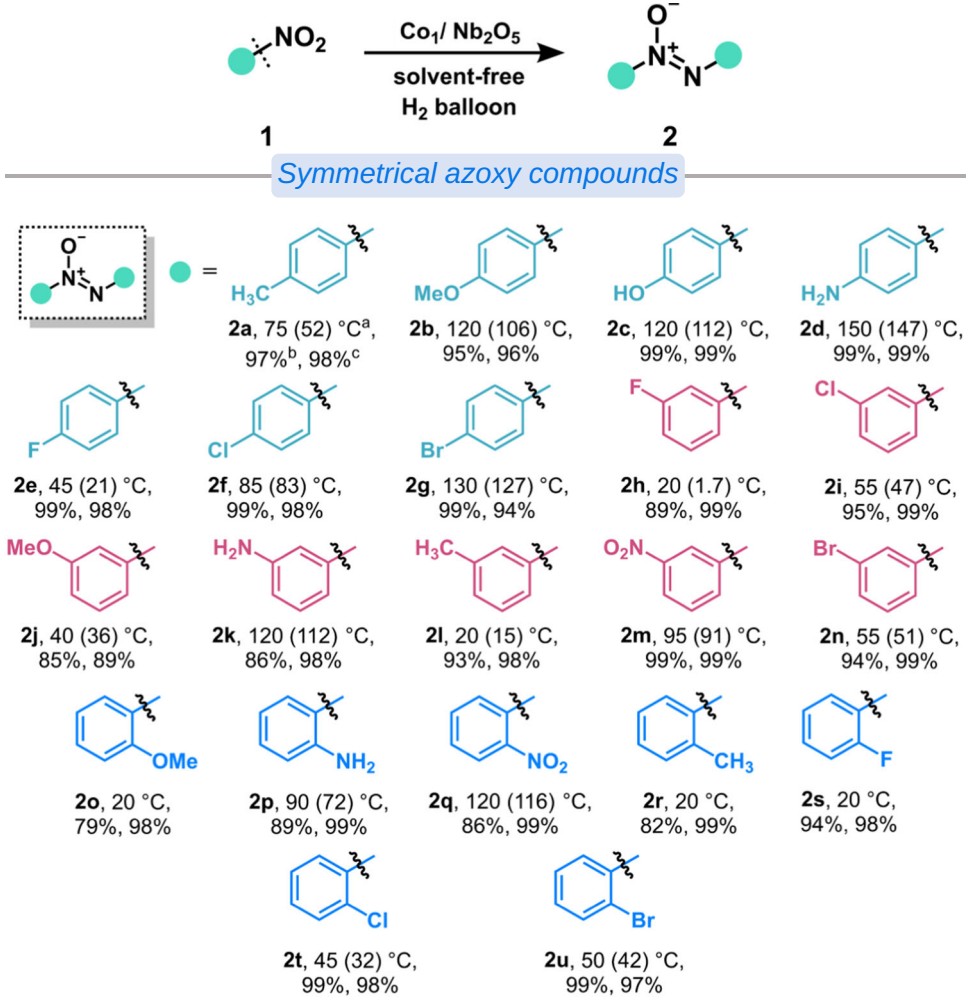

**Fig. 5 | Substrate scope of the solvent-free selective hydrogenation of nitroaromatics to symmetrical azoxy compounds.** [a] Reaction temperatures and melting points of substrates in parentheses; [b] Conversion; [c] Selectivity.

electronic coupling between Co atoms and adjacent coordinating oxygen atoms in $Nb_2O_5$ prevents the full hydrogenation of nitroarene, giving rise to a high selectivity towards azoxybenzene. Overall, DFT calculations provide solid evidence that $Co_1/Nb_2O_5$, with its unique interface and electronic properties and strong electronic metal-support interactions, can significantly affect adsorption characteristics and activation ability with reactants, thus lowering the energy barriers and facilitating the formation of azoxy compounds, which ensures prominent catalytic activity and selectivity.

## Discussion

In conclusion, we report on a facile strategy to access an efficient heterogeneous catalyst with atomically dispersed Co atoms over $Nb_2O_5$ nanomeshes. AC HAADF-STEM, CO-DRIFTS, XAFS, and XPS characterizations reveal that these isolated Co atoms are positively charged and coordinated with the neighboring oxygen atoms. This $Co_1/Nb_2O_5$ catalyst exhibits exceptional catalytic efficiency in solvent-free hydrogenation of nitrobenzene to give azoxybenzene, superior to that of reported catalysts. In addition, $Co_1/Nb_2O_5$ successfully promoted the solvent-free hydrogenation coupling of a broad range of nitroaromatics into the desired azoxy compounds with high efficiency. More importantly, excellent recyclability and reaction scale tolerance were demonstrated. Theoretical calculations elucidate that the unique electronic properties and strong electronic metal-support interactions of catalytically active Co sites

in $Co_1/Nb_2O_5$ have a substantial influence on the reaction pathways and energy barriers. Our findings underscore the great potential of this synthetic strategy for designing high-performance catalysts and provide insights into the structure-performance relationship for industrially important catalytic reactions.

## Methods

### Synthesis of $Nb_2O_5$

In a typical synthesis, 1.6 mmol of ammonium niobate oxalate hydrate, 16 mmol of melamine, and 40 mmol of ammonium chloride were dissolved in 40 ml of ethanol and stirred for 12 h. The mixture was then washed with ethanol and vacuum-dried at 80 °C. Subsequently, the dried powder was transferred to a tube furnace and heated at 550 °C in air for 4 h with a heating rate of 2.5 °C min$^{-1}$. After cooling to room temperature, the white-colored $Nb_2O_5$ nanomeshes were obtained.

### Synthesis of $Co_1/Nb_2O_5$

In a typical synthesis, 0.4 g of as-prepared $Nb_2O_5$ was subjected to an incipient wetness impregnation method with 600 μl cobalt(II) acetate ethanol solution (20 mg/ml), followed by an infrared lamp drying step ($Co^{2+}@Nb_2O_5$). The dried powder was sealed in an argon-filled glass vial and microwave-treated at 800 W for 10 s (Microwave Oven, Galanz) to obtain $Co_1/Nb_2O_5$. The metal loading in the catalyst was determined to be 0.42 wt%.

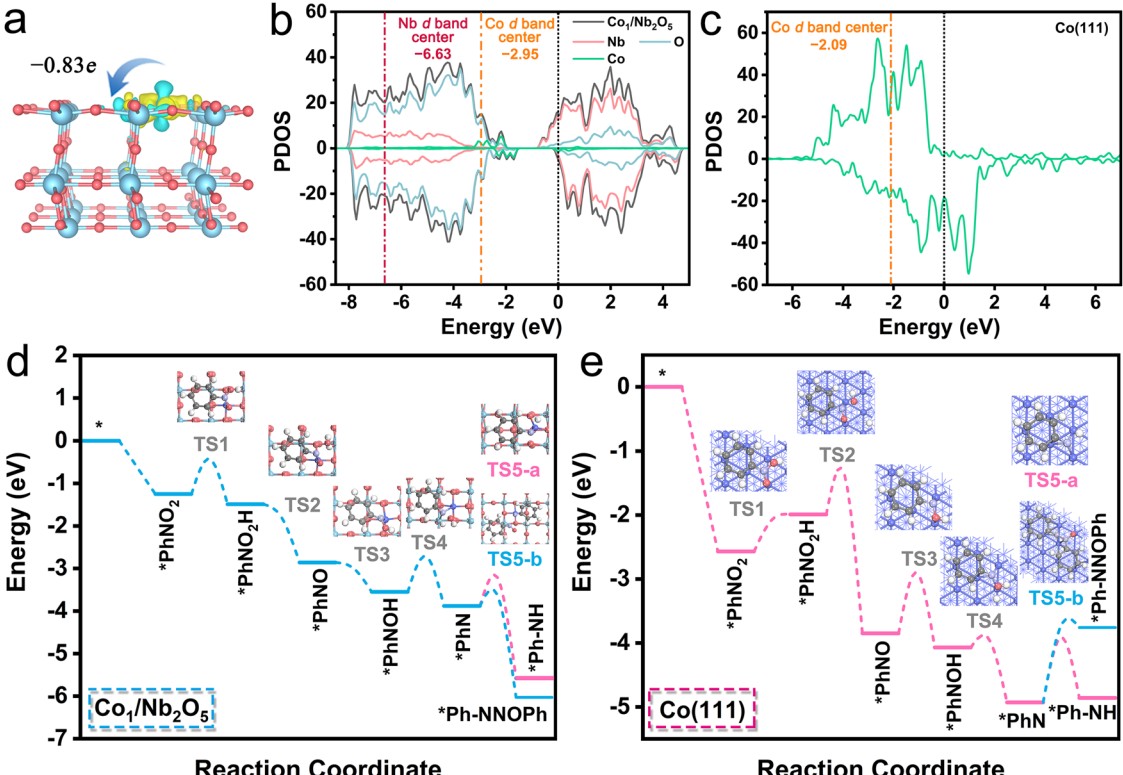

**Fig. 6 | DFT calculations. a** Bader charge and charge density difference of Co₁/Nb₂O₅. The isosurface level is 0.004 eÅ⁻³. The PDOS and *d* band centers of (**b**) Co₁/Nb₂O₅ and (**c**) Co(111). Energy profiles of solvent-free selective hydrogenation of nitrobenzene over (**d**) Co₁/Nb₂O₅ and (**e**) Co(111). Inset: the corresponding transition state configurations.

## Synthesis of Co NPs/Nb₂O₅

The preparation method was the same as that of Co₁/Nb₂O₅, except 1.3 ml of cobalt(II) acetate ethanol solution (100 mg/ml) was used. The metal loading in the catalyst was determined to be 5.12 wt%.

## Catalytic evaluation

The solvent-free selective hydrogenation of nitroaromatics on Co₁/Nb₂O₅ was evaluated under atmosphere pressure. Typically, 20 mmol of nitrobenzene and 100 mg of Co₁/Nb₂O₅ (with a molar ratio of 2800:1) were introduced into 25 ml of a round bottom flask connected with a balloon filled with H₂. The catalytic reaction was performed at 20 °C. After the reaction, 50 μl of the resultant mixture was added to 2 ml of ethyl acetate before centrifugation. The corresponding organic compounds were extracted and analyzed by gas chromatography (GC, Techcomp GC-7980) equipped with an HP-5 capillary column and a flame ionization detector. The qualitative analysis was performed by gas chromatography-mass spectrometry (GC-MS, 7890 and 5975 C, Agilent). The used catalyst was separated from the reaction mixture by centrifugation and washed with ethanol before being vacuum-dried at 60 °C for the next catalytic cycle.

For the reaction with solvents, 10 mg of Co₁/Nb₂O₅, 2 mmol of nitrobenzene, and 5 ml of mixed solvents (tetrahydrofuran: H₂O = 4:1, v:v) were added into a 25 ml of Schlenk glass vessel tube with a molar ratio of nitrobenzene: Co is 2800:1. The catalytic reaction was performed at 20 °C under H₂ atmosphere.

The turnover frequency (TOF) values of the catalysts were determined below 10% conversion of the substrate and based on exposed Co atoms. The conversion, selectivity, yield, and TOF are defined as follows:

$$\text{Conversion} = \frac{\text{mole of reacted nitrobenzene}}{\text{mole of nitrobenzene fed}} \times 100\% \quad (1)$$

$$\text{Selectivity} = \frac{\text{mole of azoxybenzene formed}}{\text{mole of nitrobenzene reacted}} \times 100\% \quad (2)$$

$$\text{TOF} = \frac{\text{mole of converted nitrobenzene}}{\text{mole of exposed cobalt atoms} \times \text{reaction time}} \times 100\% \quad (3)$$

$$\text{Yield} = \frac{\text{mole of azoxy compounds} \times 2}{\text{mole of nitrobenzene}} \times 100\% \quad (4)$$

## Data availability

The data supporting the findings of this work are available within the article and Supplementary Information. All data are available from the corresponding authors upon request.

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

## Acknowledgements

This work was supported by the Outstanding Youth Project of the Natural Science Foundation of Heilongjiang Province (YQ2022B002), the Scientific Research Foundation for Returned Scholars of Heilongjiang Province of China (719900091), and the Heilongjiang Touyan Innovation Team Program. Y.W. acknowledges financial supports from the Natural Science Foundation of China (22203044) and the Jiangsu Specially Appointed Professor Plan. The authors thank beamline 1W1B of Beijing Synchrotron Radiation Facility (BSRF) in Beijing and beamline of MCD-A and MCD-B (Soochow Beamline for Energy Materials) of National Synchrotron Radiation Laboratory in Hefei for XAS measurements. The authors also acknowledge the Catalysis and Surface Science Endstation at beamline BL11U for support in ultraviolet photoemission spectroscopy (UPS) measurements.

## Author contributions

Z.L. conceived the idea, supervised the project, and wrote the paper. Xi.L. synthesized the catalysts, performed the catalytic reactions, and analyzed the data. Co.G. and Y.W. conducted the DFT calculations. S.J., H.L., Ch.G., Xu.L., and S.X. assisted with the material synthesis and characterizations. S.J. and H.L. performed the in-situ FT-IR measurements and analyzed the data. B.L. and W.W. performed $H_2$-TPR, $H_2$-TPD, $H_2$-$D_2$ exchange, and TGA measurements. C.W. and W.Y. helped with the analysis of the XAFS spectra. J.H.H. contributed to the discussion on the experiment. All authors contributed to the overall scientific interpretation and edited the manuscript.

## Competing interests

The authors declare no competing interests.
