## [Peer Review File · Nature Communications]

Solvent-free selective hydrogenation of nitroaromatics to azoxy compounds over Co single atoms decorated on Nb₂O₅ nanomeshesREVIEWER COMMENTS

Reviewer #1 (Remarks to the Author):

This manuscript reported a highly efficient Co1/Nb2O5 catalyst for selective transfer hydrogenation coupling of nitrobenzene to azoxybenzene. Impressively, it exhibited superior high activity and selectivity, preponderant to the state-of-the-art catalysts in literature. Although the results are very interesting, some key issues are needed to be addressed before it can be accepted by Nat. Commun. Therefore, I recommend major revision at the current version.

1. One of the biggest problems in this manuscript is the reaction mechanism. How and where does H₂ dissociation? The authors didn't consider it at all in the DFT calculations. Moreover, only thermodynamic was calculated, how about the kinetic dynamic during the processes?
2. It seems the Nb₂O₅ support played the very important role for Co single atoms in this reaction. So how about Co single atoms anchored on other oxides supports with the O-coordination structure, and even Co single atoms with different coordination structures, e.g. Co₁-Nx..... If the activity or selectivity has a big difference, what's the reason?
3. The exactly position and coordination structure of Co single atoms are not presented clearly in the manuscript. From the AC HAADF-STEM images, it seems that Co single atoms replace the positions of Nb atoms of the Nb₂O₅ support. However, Co single atoms are not in the corresponding positions in the XAFS analysis model and DFT calculation model.
4. In Supplementary Fig. 17, Nb₂O₅ support exhibited nearly the same activity to that of Co₁/Nb₂O₅ catalyst? Please confirm it. In addition, Nb₂O₅ support alone also has some activity, while the selectivity is not good. And then why no side products with Co₁/Nb₂O₅?
5. The high selectivity of Co₁/Nb₂O₅ is might related to the moderate reaction conditions (20 °C, H₂ balloon). How about the performances under pressure and elevated temperature?
6. It can be seen from Fig. 4 that only ~ 80% conversion for some substrates, what's the reason?
7. Co k-edge of the re-used Co₁/Nb₂O₅ should be test, not L-edge.

Reviewer #2 (Remarks to the Author):

The authors presented an intriguing finding, demonstrating the excellent performance of Co₁/Nb₂O₅ in the selective hydrogenation coupling of nitrobenzene to azoxybenzene. However, before publication, certain aspects need to be addressed, particularly regarding the discussion on the reaction route over Co₁/Nb₂O₅ and Co NPs/Nb₂O₅.

(1) The referee suggests replacing "transfer hydrogenation" with "hydrogenation" since H₂ is the hydrogen source for the selective hydrogenation coupling of nitrobenzene.

(2) It is important to examine the atomic dispersion using CO-probe Fourier transform infrared spectroscopy.

(3) The discussion about Raman spectra is wrong. After Co deposition, the intensities of peaks at 120 cm^{-1} in Co1/Nb2O5 and Co NPs/Nb2O5 are enhanced, while the peaks at 230 cm^{-1} are weakened.

(4) The reaction routes have been discussed for Co1/Nb2O5 and Co NPs/Nb2O5. Fig. 3d shows that azoxybenzene is the intermediate product of nitrobenzene hydrogenation to aniline under solvent-free conditions over Co NPs/Nb2O5. The difference in catalytic performance between Co1/Nb2O5 and Co NPs/Nb2O5 lies in the activity for the hydrogenation of azoxybenzene to aniline. However, there is a contradiction between the DFT calculation results and the experimental findings. The DFT calculations suggest that the production of Ph-NNOPh is difficult over Co NPs/Nb2O5, whereas experimental results indicate its high selectivity in Fig 3d. Moreover, it is worth mentioning that the authors only considered the Co site in the calculations without taking into account the active site on the niobium oxide. In the reaction route shown in Figure 3f, niobium oxide may catalyze the dehydration reaction in step 3, explaining the high selectivity of Ph-NNOPh over Co NPs/Nb2O5.

Reviewer #3 (Remarks to the Author):

In this manuscript, the authors present a solvent-free hydrogenation transfer strategy employing single supported Co single-atom catalysts (SACs) to achieve high activity and selectivity in the coupling of nitroaromatics to azoxy compounds. The authors have developed a simple and reliable method for synthesizing atomically dispersed Co atoms anchored on Nb2O5. The resulting Co1/Nb2O5 material exhibits a robust electronic coupling effect between the Co atoms and the support atoms. Through detailed characterization, the authors have identified the interaction between isolated Co atoms and coordinated oxygen atoms as the key factor behind the efficient solvent-free transfer hydrogenation coupling of various nitroaromatics into desired azoxy compounds. The interesting findings will make important contribution to the field of fast-growing single-atom catalysis. The manuscript is deserving of publication in nature communications, pending some revisions.

1. The manuscript discusses highly efficient catalysts in a solvent-free system. The authors need to clarify the importance of introducing a solvent-free method in hydrogenation reactions and highlight the advantages/benefits of using atomically dispersed Co SACs in the absence of solvent molecules. This will provide a more comprehensive perspective on the catalytic process.
2. The authors verified the highly catalytic performance of Co1/Nb2O5 using solvent-free method. However, it's important to emphasize the differences between using solvent and not using solvent, particularly when both methods achieve similar effect (99% conversion and 99% selectivity). Additionally, the presence of solvent can influence product selectivity, which may not be achievable through the solvent-free method. The impact of solvent molecules on the reaction's side effects could be discussed.
3. The difference of the molar ratio of Co to nitrobenzene when comparing the hydrogenation catalytic performance with and without solvent participation could cause an inaccurate comparison of TOF. To address this, it is essential to maintain consistent reaction conditions.

4. The author claimed that the lower activation energy of Co1/Nb2O5 compared to Co NPs/Nb2O5 (Fig. 3g) suggests enhanced catalytic activity. However, it remains unclear why Nb2O5, despite having the lowest activation energy, exhibits the lowest reaction activity among the three samples (Fig. 3b)? Since there should be no diffusion differences between the samples. The corresponding explanation should be provided.

5. It is noticed that the resulting products exhibit a certain degree of symmetry. Did the author consider synthesizing the products using different nitroaromatic compounds to explore potential variations?

6. Some relevant papers shall be cited, eg. Advanced Materials, 2008471, 2021, National Science Review 10 (1), nwac100, 2023

7. The Co-O bond length in Co1/Nb2O5 is noticeably shorter at 1.54 Å compared to 1.75 Å in CoO (Fig. 2h). Does this correspond to the simulated bond length in DFT calculations?

8. How do the authors confirm the products? The corresponding GC-MS data or NMR spectroscopy of the products are suggested to be provided.

No. NCOMMS-23-33366

Title: Solvent-free selective hydrogenation of nitroaromatics to azoxy compounds over Co single atoms decorated on Nb₂O₅ nanomeshes

Author(s): Zhijun Li^{1,*} Xiaowen Lu,^{1,†} Cong Guo,^{1,†} Siqu Ji,¹ Hongxue Liu,¹ Chunmin Guo,¹ Xue Lu,¹ Chao Wang,³ Wensheng Yan,³ Bingyu Liu,⁴ Wei Wu,⁴ J. Hugh Horton,^{1,5} Shixuan Xin,¹ Yu Wang^{2,*}

Journal: *Nature Communications*

We would like to thank all the reviewers and editors for your time and efforts in evaluating our manuscript. We have made all of the requested changes, which are listed point-by-point in this letter (*Italic, blue*).

Response to Referee

Reviewer #1: This manuscript reported a highly efficient Co1/Nb2O5 catalyst for selective transfer hydrogenation coupling of nitrobenzene to azoxybenzene. Impressively, it exhibited superior high activity and selectivity, preponderant to the state-of-the-art catalysts in literature. Although the results are very interesting, some key issues are needed to be addressed before it can be accepted by Nat. Common. Therefore, I recommend major revision at the current version.

1. One of the biggest problems in this manuscript is the reaction mechanism. How and where does H₂ dissociation? The authors didn't consider it at all in the DFT calculations. Moreover, only thermodynamic was calculated, how about the kinetic dynamic during the processes?

*Response: Thanks for the reviewer's insightful comment. We agree with the reviewer that H₂ dissociation and kinetics are essential for demonstrating the reaction mechanism. Previous studies demonstrated that H₂ tends to undergo heterolytic dissociation on Nb₂O₅(001) with a kinetic energy barrier (E_b) of 0.88 eV (ACS Catal. 2022, 12, 4806–4812), and on the Co (111) surface, the dissociation barrier is 0.03 eV (Int. J. Hydrog. Energy 2018, 43, 5576-5590). In this regard, we further performed additional calculations to investigate the kinetics during the processes. It can be found from Figure 5d,e that for the first four hydrogenation steps (*PhNO₂ → *PhNO₂H → *PhNO → *PhNOH → *PhN), the major kinetic barrier of Co₁/Nb₂O₅ occurs in the hydrogenation of *PhNOH to *PhN (0.84 eV), whereas that of Co(111) is the hydrogenation of *PhNO to *PhNOH (0.96 eV). As mentioned in the main text, *PhN can couple with *PhNO to form Ph-NNOPh or hydrogenate to *PhN, thereby determining the product selectivity. Remarkably, Co₁/Nb₂O₅ exhibits a favorable E_b of 0.22 eV for facilitating the coupling of *PhN and *PhNO to Ph-NNOPh, which is lower than that of the competing *PhNH formation (0.65 eV). By contrast, the hydrogenation of *PhN to *PhNH on Co (111) requires overcoming a moderate E_b of 1.02 eV, whereas the barrier for the Ph-NNOPh generation is 1.21 eV. These kinetic results are consistent with the experimental observation.*

The above discussion has been added to the revised Manuscript and Supplementary Information. Please see Pages 10 to 11 of the revised manuscript “We also performed a kinetic analysis to gain more insights into the reactivity of Co₁/Nb₂O₅ and Co(111). The corresponding intermediates configurations are displayed in Supplementary Figs. 51 and 52. Please note that H₂ can undergo heterolytic dissociation on the Nb₂O₅(001) surface with a kinetic barrier (E_b) of 0.88 eV to provide

the hydrogen source⁶⁰, and on the Co(111) surface, the dissociation barrier is 0.03 eV⁶¹. According to our calculations, for the first four hydrogenation steps, the major E_b of Co₁/Nb₂O₅ occurs in the conversion of *PhNOH to *PhN (0.84 eV), and that of Co(111) is the conversion of *PhNO to *PhNOH (0.95 eV). Specifically, Co₁/Nb₂O₅ exhibits a favorable E_b of 0.22 eV for facilitating the further coupling of *PhN and *PhNO to Ph-NNOPh, which is lower than that of the competing *PhNH formation (0.65 eV). By contrast, the hydrogenation of *PhN to *PhNH on Co(111) requires overcoming a moderate E_b of 1.02 eV, whereas the barrier for the Ph-NNOPh generation is 1.21 eV.”. And Page 5 of the revised Supplementary Information: “The transition state (TS) was determined using the double-ended surface walking (DESW) and constrained Broyden dimer (CBD) approaches, as implemented in the LASP software^{7,8}; the methods can establish a low-energy pathway linking two minima even without iterative optimization of the pathway, from which the TS can be located readily. All transition states were verified by vibrational frequency calculations (only one imaginary frequency).”.

Fig. 5. DFT calculations. (a) Bader charge and charge density difference of Co₁/Nb₂O₅. The isosurface level is 0.004 eÅ⁻³. The PDOS and d band centers of (b) Co₁/Nb₂O₅ and (c) Co(111). Energy profiles of transfer hydrogenation coupling of nitrobenzene to azoxybenzene over (d) Co₁/Nb₂O₅ and (e) Co(111). Inset: the corresponding transition state configurations.

2. It seems the Nb₂O₅ support played the very important role for Co single atoms in this reaction. So how about Co single atoms anchored on other oxides supports with the O-coordination structure, and even Co single atoms with different coordination structures, e.g. Co₁-Nx..... If the activity or selectivity has a big difference, what's the reason?

Response: We thank the reviewer for this important comment. We have employed the same synthetic method and fabricated two types of oxide-supported Co single atom catalysts, being Co₁/V₂O₅ and Co₁/MgO, respectively. In addition, we prepared a nitrogen-doped carbon

supported Co single atom catalyst ($\text{Co}_1/\text{N-C}$) as per your request. We found that there is indeed a big difference in catalytic activity between them. We think this can be ascribed to the unique coordination environments, electronic structures, and electronic metal-support interactions of catalytically active Co sites in $\text{Co}_1/\text{Nb}_2\text{O}_5$ than in other catalysts to significantly influence the reaction pathways and energy barriers. Please see Supplementary Figs. 20-22 in Supplementary Information, Fig. 3c and Page 8 of the revised manuscript “Co single atoms anchored on other oxides by the same synthetic method with O-coordination structures were also prepared, being Co_1/MgO and $\text{Co}_1/\text{V}_2\text{O}_5$, respectively (Supplementary Figs. 20, 21, and Table 2). A nitrogen-doped carbon support with Co single atoms ($\text{Co}_1/\text{N-C}$) was fabricated to represent different coordination environments, for example, Co-N coordination (Supplementary Fig. 22 and Table 2). Remarkably, an exceptional TOF value of 40377 h^{-1} for $\text{Co}_1/\text{Nb}_2\text{O}_5$ was determined, significantly higher than other control samples (Fig. 3c). Poor catalytic activity and selectivity of Co_1/MgO , $\text{Co}_1/\text{V}_2\text{O}_5$, and $\text{Co}_1/\text{N-C}$ are observed (Supplementary Fig. 23). This might result from the unique coordination environments, electronic structures, and electronic metal-support interactions of catalytically active Co sites in $\text{Co}_1/\text{Nb}_2\text{O}_5$ that influence the reaction pathways and energy barriers.”.

Supplementary Fig. 20 Characterization of the as-prepared Co_1/MgO . (a) XANES spectra at Co K-edge. (g) Fourier transformed k^3 -weighted Co K-edge of EXAFS spectra. (c) EXAFS fittings in R space.

Supplementary Fig. 21 Characterization of the as-prepared $\text{Co}_1/\text{V}_2\text{O}_5$. (a) XANES spectra at Co K-edge. (g) Fourier transformed k^3 -weighted Co K-edge of EXAFS spectra. (c) EXAFS fittings in R space.

Supplementary Fig. 22 Characterization of the as-prepared $\text{Co}_1/\text{N}-\text{C}$. (a) XANES spectra at Co K-edge. (g) Fourier transformed k^3 -weighted Co K-edge of EXAFS spectra. (c) EXAFS fittings in R space.

Fig. 3. Catalytic performances of $\text{Co}_1/\text{Nb}_2\text{O}_5$ in the solvent-free selective hydrogenation of nitrobenzene to azoxybenzene. Conversion and selectivity of (a) $\text{Co}_1/\text{Nb}_2\text{O}_5$ and (b) $\text{Co NPs}/\text{Nb}_2\text{O}_5$. (c) The corresponding TOF values of different samples. (d) Arrhenius plots and E_a values. (e) Recycling results. (f) The proposed reaction routes. In situ DRIFTS spectra recorded during the hydrogenation of nitrobenzene over (g) $\text{Co}_1/\text{Nb}_2\text{O}_5$ and (h) $\text{Co NPs}/\text{Nb}_2\text{O}_5$. (NB: nitrobenzene; NSB: nitrosobenzene; PHA: phenylhydroxylamine; AB: azobenzene; AOB: azoxybenzene; AN: aniline)

3. The exactly position and coordination structure of Co single atoms are not presented clearly in the manuscript. From the AC HAADF-STEM images, it seems that Co single atoms replace the positions of Nb atoms of the Nb₂O₅ support. However, Co single atoms are not in the corresponding positions in the XAFS analysis model and DFT calculation model.

Response: Thanks for this insightful comment and amendments have been added to the manuscript. Please see Supplementary Fig. 3 in Supplementary Information, Figure 1f (Page 19) and Page 4 of the revised manuscript “Additionally, the enlarged area provides preliminary evidence for the presence of isolated Co atoms over the Nb₂O₅ surface (Fig. 1f inset).”.

Fig. 1. The synthesis strategy and characterizations of Co₁/Nb₂O₅. (a) Schematic illustration of the synthesis strategy. (b) SEM image. (c) TEM image. (d) HR-TEM image (the inset is an AFM image showing the height profile). (e) AC HAADF-STEM image (the inset is SAED pattern). (f) AC HAADF-STEM image at high magnification. The red, blue, and purple atoms represent O, Nb, and Co, respectively. (g) EDS mapping images.

4. In Supplementary Fig. 17, Nb₂O₅ support exhibited nearly the same activity to that of Co₁/Nb₂O₅ catalyst? Please confirm it. In addition, Nb₂O₅ support alone also has some

activity, while the selectivity is not good. And then why no side products with Co1/Nb2O5?

Response: Thanks for the reviewer's important comment. As shown in Supplementary Fig. 24, the activity of Nb₂O₅ was indeed significantly lower than that of Co₁/Nb₂O₅. The reason for the high selectivity can be found in the DFT sections in Pages 10 and 11 of the revised manuscript.

Supplementary Fig. 24 Kinetic results of the solvent-free selective hydrogenation of nitrobenzene to azoxybenzene over (a) Nb₂O₅, (b) Co₁/Nb₂O₅, and (c) Co NPs/Nb₂O₅ at different temperatures.

5. The high selectivity of Co1/Nb2O5 is might related to the moderate reaction conditions (20 °C, H2 balloon). How about the performances under pressure and elevated temperature?

Response: We thank the reviewer for this important point. We have carried out the corresponding catalytic performance under different pressures and elevated temperatures. We found that increasing the H₂ pressure and temperature result in higher reaction rates but slightly lowered selectivity for the target product of azobenzene. Therefore, our mild experimental conditions are supposed to be a better choice because of the high reaction activity and exceptional selectivity.

Table R1. Catalytic Performance of Co₁/Nb₂O₅ under higher pressures and elevated temperatures.

Entry	P _{H2} (MPa)	T (°C)	Time (min)	Conv. (%)	Sel. (%)		
					azoxybenzene	azobenzene	aniline
1	0.5	80	25	>99	99	1	–
2	1	80	20	>99	97	–	3
3	2	80	15	>99	96	1	3
5	1	120	15	>99	98	1	1
6	2	120	10	>99	95	2	3

6. It can be seen from Fig. 4 that only ~ 80% conversion for some substrates, what's the

reason?

Response: Thanks for the reviewer's comment and the corresponding amendments have been done. Please see Page 9 of the revised manuscript "Note that methyl and methoxy groups at either ortho- or meta-positions (2r, 2o, 2l, 2j) underwent smooth conversion to afford the desired products with slightly lower efficacy compared to the more sterically accessible ones at para-position (2a, 2b). This might be due to the steric hindrance effect of the substituent groups that unfavor the coupling reactions."

7. Co k-edge of the re-used Co₁/Nb₂O₅ should be test, not L-edge.

Response: We have added Co k-edge results of the spent Co₁/Nb₂O₅. Please see Supplementary Fig. 26 and Table S2 of the revised Supplementary Information and Page 8 of the revised manuscript "Importantly, the EXAFS spectrum of spent Co₁/Nb₂O₅ catalyst (Supplementary Fig. 26) demonstrates that the dispersion and coordination environment of Co atoms do not show obvious variations after 10 cycles. In addition, Co L-edge and O K-edge NEXAFS (Supplementary Fig. 27) reveal that no substantial electronic structural changes were observed. These results imply a strong metal-support interaction in the Co₁/Nb₂O₅ catalyst with high stability."

Supplementary Fig. 26 Characterization of the spent Co₁/Nb₂O₅ after recycling tests. (a) XANES spectra at Co K-edge. (g) Fourier transformed k^3 -weighted Co K-edge of EXAFS spectra. (c) EXAFS fittings in R space.

Supplementary Table 2. Structural parameters of Co₁/Nb₂O₅ extracted from the EXAFS fitting with reference of Co foil at the Co K-edge. ($S_0^2=0.82$)

Sample	Path	N	R (Å)	σ^2 (10^{-3}Å^2)	ΔE_0 (eV)	R factor
--------	------	---	-------	------------------------------------	-------------------	----------

Co foil	Co-Co	12	2.49	2.7	6.1	0.001
Co₁/Nb₂O₅ (fresh)	Co-O	4.6	2.00	3.0	-4.6	0.001
Co₁/Nb₂O₅ (spent)	Co-O	4.5	2.01	3.0	4.9	0.010
	Co-O	4.1	2.11	6.8	2.8	
Co₁/MgO	Co-O-Mg	5.9	3.01	9.5	-6.3	0.011
Co₁/V₂O₅	Co-O	5.2	2.08	5.1	-0.1	0.010
Co₁/N-C	Co-N	3.8	1.99	5.4	6.3	0.006

N: coordination numbers; R: bond distance; σ^2 : Debye-Waller factors; ΔE_0 : the inner potential correction. R factor: goodness of fit.

Reviewer #2: The authors presented an intriguing finding, demonstrating the excellent performance of Co₁/Nb₂O₅ in the selective hydrogenation coupling of nitrobenzene to azoxybenzene. However, before publication, certain aspects need to be addressed, particularly regarding the discussion on the reaction route over Co₁/Nb₂O₅ and Co NPs/Nb₂O₅.

1. The referee suggests replacing "transfer hydrogenation" with "hydrogenation" since H₂ is the hydrogen source for the selective hydrogenation coupling of nitrobenzene.

Response: Thanks for this insightful suggestion and the corresponding corrections have been done throughout the revised version of manuscript and Supplementary Information.

2. It is important to examine the atomic dispersion using CO-probe Fourier transform infrared spectroscopy.

Response: We thank the reviewer for this useful comment and the corresponding CO-FTIR results have been added to the revised manuscript. Please see Page 5 of the revised manuscript, Fig. 2d and Supplementary Fig. 6 "In situ CO-diffuse reflectance infrared Fourier transform spectroscopy (CO-DRIFTS) was performed to investigate atomic structures of Co species in Co₁/Nb₂O₅ and Co NPs/Nb₂O₅ (Fig. 2d). The broad bands around 2171 cm⁻¹ and 2115 cm⁻¹ are ascribed to residual gas-phase CO molecules^{47,48}. For Co₁/Nb₂O₅, a pair of CO adsorption peaks were observed at 2089 cm⁻¹ and 2067 cm⁻¹, respectively, which could be assigned to linearly adsorbed CO on positively charged Co species^{49,50}. The absence of other peaks excludes the presence of Co multiatomic species in Co₁/Nb₂O₅. In the case of Co NPs/Nb₂O₅, two peaks appeared at 2015 cm⁻¹ and 1984 cm⁻¹, respectively, which are associated with the CO adsorbing on metallic Co in linear and bridge form (Supplementary Fig. 6)⁵¹."

Fig. 2 Structural characterizations of as-prepared $\text{Co}_1/\text{Nb}_2\text{O}_5$. (a) XRD patterns. (b) Raman spectra. (c) XPS Co 2p spectra. (d) In situ CO-DRIFTS of $\text{Co}_1/\text{Nb}_2\text{O}_5$. (e) O K-edge spectra. (f) XANES spectra at Co K-edge. (g) Fourier transformed k^3 -weighted Co K-edge of EXAFS spectra. (h) EXAFS fitting in R space (inset is the model of Co_1O_4). (i) 3D contour WT-EXAFS plot.

Supplementary Fig. 6 In situ CO-DRIFTS of Co NPs/Nb₂O₅.

3. The discussion about Raman spectra is wrong. After Co deposition, the intensities of peaks at 120 cm⁻¹ in Co₁/Nb₂O₅ and Co NPs/Nb₂O₅ are enhanced, while the peaks at 230 cm⁻¹ are weakened.

Response: Thanks very much for the reviewer's comment and the corrections have been done. Please see Page 4 "After Co deposition, the intensities of peaks at 120 cm⁻¹ in Co₁/Nb₂O₅ and Co NPs/Nb₂O₅ are enhanced, while the peaks at 230 cm⁻¹ are weakened."

4. The reaction routes have been discussed for Co₁/Nb₂O₅ and Co NPs/Nb₂O₅. Fig. 3d shows that azoxybenzene is the intermediate product of nitrobenzene hydrogenation to aniline under solvent-free conditions over Co NPs/Nb₂O₅. The difference in catalytic performance between Co₁/Nb₂O₅ and Co NPs/Nb₂O₅ lies in the activity for the hydrogenation of azoxybenzene to aniline. However, there is a contradiction between the DFT calculation results and the experimental findings. The DFT calculations suggest that the production of Ph-NNOPh is difficult over Co NPs/Nb₂O₅, whereas experimental results indicate its high selectivity in Fig 3d. Moreover, it is worth mentioning that the authors only considered the Co site in the calculations without taking into account the active site on the niobium oxide. In the reaction route shown in Figure 3f, niobium oxide may catalyze the dehydration reaction in step 3, explaining the high selectivity of Ph-NNOPh over Co NPs/Nb₂O₅.

Response: We thank the reviewer very much for the important comments. We have checked all the graphs and found the mistakes in original Fig. 3d. In the revised manuscript, we have corrected this point and this time the experimental results are in good agreement with the DFT calculations. Please see Fig. 3a,b,and d in the revised manuscript.

*We also thank the reviewer for reminding us that niobium oxide may also be essential for the dehydration reaction. Indeed, we found that the active sites of niobium oxide can promote the coupling of *PhNO and *PhN on Co₁/Nb₂O₅ (Fig. 5d). According to our kinetic analysis, the Nb site of Co₁/Nb₂O₅ exhibits favorable binding to *PhNO, and then the *PhNO can couple with the *PhN (adsorbed on the Co₁ site of Co₁/Nb₂O₅) to form Ph-NNOPh, resulting in a small kinetic barrier of 0.22 eV. By contrast, the corresponding kinetic barrier of Co(III) is as high as 1.21 eV. Please see Page 11 of the revised manuscript "Please note that the active sites of the niobium oxide component are essential for the dehydration reaction because the Nb site of Co₁/Nb₂O₅ exhibits favorable binding to *PhNO, which promotes the coupling of *PhNO and *PhN."*

Fig. 3. Catalytic performances of Co₁/Nb₂O₅ in the solvent-free selective hydrogenation of nitrobenzene to azoxybenzene. Conversion and selectivity of (a) Co₁/Nb₂O₅ and (b) Co NPs/Nb₂O₅. (c) The corresponding TOF values of different samples. (d) Arrhenius plots and E_a values. (e) Recycling results. (f) The proposed reaction routes. In situ DRIFTS spectra recorded during the hydrogenation of nitrobenzene over (g) Co₁/Nb₂O₅ and (h) Co NPs/Nb₂O₅. (NB: nitrobenzene; NSB: nitrosobenzene; PHA: phenylhydroxylamine; AB: azobenzene; AOB: azoxybenzene; AN: aniline)

Reviewer #3: In this manuscript, the authors present a solvent-free hydrogenation transfer strategy employing single supported Co single-atom catalysts (SACs) to achieve high activity and selectivity in the coupling of nitroaromatics to azoxy compounds. The authors have developed a simple and reliable method for synthesizing atomically dispersed Co atoms anchored on Nb₂O₅. The resulting Co₁/Nb₂O₅ material exhibits a robust electronic coupling effect between the Co atoms and the support atoms. Through detailed characterization, the authors have identified the interaction between isolated Co atoms and coordinated oxygen atoms as the key factor behind the efficient solvent-free transfer hydrogenation coupling of various nitroaromatics into desired azoxy compounds. The interesting findings will make important contribution to the field of fast-growing single-atom catalysis. The manuscript is deserving of publication in nature communications, pending some revisions.

1. The manuscript discusses highly efficient catalysts in a solvent-free system. The authors need to clarify the importance of introducing a solvent-free method in hydrogenation reactions and highlight the advantages/benefits of using atomically dispersed Co SACs in the absence of solvent molecules. This will provide a more comprehensive perspective on the catalytic process.

Response: Thanks for this important comment and corresponding amendments have been done. Please see Page 2 of the revised manuscript. "In addition to the catalytic activity-selectivity relationship, solvent waste removal also poses a formidable challenge for green chemical synthesis and energy consumption¹⁵. Encouragingly, the use of solvent-free mechanical methods (grinding or milling) for reagents mixing and activation holds many advantages, such as shortened reaction periods, mild reaction conditions, and excellent selectivity to the target products¹⁶⁻¹⁸. Therefore, there is a strong incentive to construct highly active and selective catalyst systems that do not require the use of solvent to efficiently hydrogenate nitroaromatics to the corresponding azoxy compounds using H₂^{19,20}."

2. The authors verified the highly catalytic performance of Co₁/Nb₂O₅ using solvent-free method. However, it's important to emphasize the differences between using solvent and not using solvent, particularly when both methods achieve similar effect (99% conversion and 99% selectivity). Additionally, the presence of solvent can influence product selectivity, which may not be achievable through the solvent-free method. The impact of solvent molecules on the reaction's side effects could be discussed.

Response: Thanks very much for this insightful suggestion. During the revision, we performed DFT calculations to explore the influence of solvent on the product selectivity. The solvent effect was described using the Poisson-Boltzmann implicit solvation model, in which the dielectric constant ϵ was taken as 18.36 for THF: H₂O ($v: v = 4: 1$) as a demonstration. Both the thermodynamic and kinetic results reveal that the selectivity of Co(III) and Co₁/Nb₂O₅ is the same as the solvent-free cases. For example, the kinetic analysis of the critical steps shows that the energy barrier (E_b) of Ph-NNOPh formation on Co₁/Nb₂O₅ (0.49 eV) is lower than that of the competing *PhN formation (0.64 eV); for Co(III), the *PhN formation ($E_b = 0.92$ eV) is more energetically favorable than Ph-NNOPh formation ($E_b = 1.20$ eV). Therefore, the solvent effect

did not change the product selectivity of $\text{Co}_1/\text{Nb}_2\text{O}_5$ and $\text{Co}(111)$, which is consistent with the experimental results.

Please see Page 10 of the revised manuscript “In addition, we explored the influence of solvent on the reaction processes and found that the solvent effect did not change the product selectivity of $\text{Co}_1/\text{Nb}_2\text{O}_5$ and $\text{Co}(111)$ (Supplementary Figs. 53 and 54). For example, after considering the solvent effect, the E_b of Ph-NNOPh formation on $\text{Co}_1/\text{Nb}_2\text{O}_5$ (0.49 eV) is still lower than that of the competing *PhN formation (0.64 eV). These results are consistent with the experimental observations.” And Page 5 of the revised Supplementary Information “The solvent effect was considered using the Poisson-Boltzmann implicit solvation model⁹, in which the dielectric constant ϵ was taken as 18.36 for THF: H_2O ($v: v = 4: 1$) as a demonstration¹⁰.”.

Supplementary Fig. 53 Energy profiles of selective hydrogenation of nitrobenzene to azoxybenzene with solvent over $\text{Co}_1/\text{Nb}_2\text{O}_5$.

Supplementary Fig. 54 Energy profiles of selective hydrogenation of nitrobenzene to azoxybenzene with solvent over $\text{Co}(111)$.

3. The difference of the molar ratio of Co to nitrobenzene when comparing the hydrogenation catalytic performance with and without solvent participation could cause an inaccurate comparison of TOF. To address this, it is essential to maintain consistent reaction conditions.

Response: Thanks for this insightful comment and we have made amendments accordingly. Please

see Supplementary Tables 3-5.

Supplementary Table 3. The effects of the reaction temperatures and solvents on the selective hydrogenation of nitrobenzene to azoxybenzene over $\text{Co}_1/\text{Nb}_2\text{O}_5$.

Entry	Solvent	T (°C)	Time (h)	Conv. (%)	Sel. (%)		
					azoxybenzene	azobenzene	aniline
1 ^a	THF/H ₂ O=4:1	20	1.5	>99	>99	–	<1
2 ^a	THF/H ₂ O=4:1	60	1	>99	98	–	2
3 ^a	Ethanol/H ₂ O=4:1	20	1.5	78	75.3	1.3	23.4
4 ^b	Toluene	20	12	<1	–	–	99
5 ^c	Acetonitrile/H ₂ O=4:1	20	8	>99	13.6	–	86.4
6 ^b	Isopropanol	20	10	<1	–	–	99

^aReaction conditions: 2 mmol nitrobenzene, 10 mg catalyst, 5 ml solvent, 1 atm H₂.

^bReaction conditions: 2 mmol nitrobenzene, 10 mg catalyst, 5 ml solvent, 1 atm H₂.

^cReaction conditions: 2 mmol nitrobenzene, 3 ml H₂O₂, 10 mg catalyst, 5 ml solvent, 1 atm H₂.

Supplementary Table 4. Reaction conditions optimization for selective hydrogenation of nitrobenzene to azoxybenzene over $\text{Co}_1/\text{Nb}_2\text{O}_5$.

Entry	Solvent	Base	Time (h)	Conv. (%)	Sel. (%)		
					azoxybenzene	azobenzene	aniline
1 ^a	THF/H ₂ O=4:1	–	1.5	>99	>99	–	<1
2 ^b	Ethanol/H ₂ O=4:1	NaOH	10	<1	–	–	99
3 ^b	Isopropanol	NaOH	10	<1	–	–	99
4 ^a	Toluene	–	12	<1	–	–	99

^aReaction conditions: 2 mmol nitrobenzene, 10 mg catalyst, 5 ml solvent, 20 °C, 1 atm H₂.

^bReaction conditions: 2 mmol nitrobenzene, 10 mg catalyst, 5 ml solvent, 0.06 mmol NaOH, 20 °C, 1 atm H₂.

Supplementary Table 5. Catalytic performance of different catalysts in the selective hydrogenation of nitrobenzene to azoxybenzene with solvent.

Catalyst	Conv. (%)	Time (min)	Sel. (%)			TOF (h ⁻¹)
			azoxybenzene	azobenzene	aniline	
Co ₁ /Nb ₂ O ₅	>99	90	>99	–	<1	11524
Co NPs/Nb ₂ O ₅	>99	110	0.8	0.2	99	4875
Nb ₂ O ₅	11.7	120	39.1	4.8	56.1	–
Co(NO ₃) ₂	19.4	90	60.5	15.8	23.7	429
CoCl ₂	9.8	90	90.6	3.8	5.6	183
Co(Ac) ₂	30.3	90	67.3	3.4	29.3	550
CoPc	8.9	90	76.5	9.7	13.8	166

Reaction conditions: 2 mmol nitrobenzene, 10 mg catalyst, 5 ml solvent (THF: H₂O=4:1), 1 atm H₂.

4. The author claimed that the lower activation energy of Co1/Nb2O5 compared to Co NPs/Nb2O5 (Fig. 3g) suggests enhanced catalytic activity. However, it remains unclear why Nb2O5, despite having the lowest activation energy, exhibits the lowest reaction activity among the three samples (Fig. 3b)? Since there should be no diffusion differences between the samples. The corresponding explanation should be provided.

Response: We thank the reviewer for pointing out this point. We have carefully checked this data and found out there was a mistake here. In the revised manuscript, the corrections have been made. Please see Fig. 3d in the revised manuscript.

Fig. 3. Catalytic performances of Co₁/Nb₂O₅ in the solvent-free selective hydrogenation of nitrobenzene to azoxybenzene. Conversion and selectivity of (a) Co₁/Nb₂O₅ and (b) Co NPs/Nb₂O₅. (c) The corresponding TOF values of different samples. (d) Arrhenius plots and E_a values. (e) Recycling results. (f) The proposed reaction routes. In situ DRIFTS spectra recorded during the hydrogenation of nitrobenzene over (g) Co₁/Nb₂O₅ and (h) Co NPs/Nb₂O₅. (NB: nitrobenzene; NSB: nitrosobenzene; PHA: phenylhydroxylamine; AB: azobenzene; AOB: azoxybenzene; AN: aniline)

5. It is noticed that the resulting products exhibit a certain degree of symmetry. Did the author consider synthesizing the products using different nitroaromatic compounds to explore potential variations?

Response: Thanks for this kind suggestion and we have tried several nitroaromatic compounds to

further explore the potential applications as per your request. We found that $\text{Co}_1/\text{Nb}_2\text{O}_5$ catalyst shows good reactivity and affords the desired unsymmetrical compounds in moderate yields in most cases. The reason is that there are competitive chemical reactions between symmetrical azoxy compounds and unsymmetrical azoxy compounds.

Please see Supplementary Fig. 45 in the revised Supplementary Information and Page 9 of the revised manuscript “In addition to symmetrical azoxy compounds, we also tried to explore the potential feasibility of synthesizing products with unsymmetrical compounds (Supplementary Fig. 45). To our delight, this $\text{Co}_1/\text{Nb}_2\text{O}_5$ catalyst shows good reactivity and affords the desired unsymmetrical compounds in moderate yields in most cases.”.

$\text{R}_1 = \text{H}, \text{R}_2 = p\text{-Cl}; t = 30 \text{ min}; \text{Yield} = 76\%$;
 $\text{R}_1 = \text{H}, \text{R}_2 = p\text{-CH}_3; t = 1 \text{ h}; \text{Yield} = 49\%$;
 $\text{R}_1 = \text{H}, \text{R}_2 = p\text{-OCH}_3; t = 1.5 \text{ h}; \text{Yield} = 24\%$;
 $\text{R}_1 = p\text{-CH}_3, \text{R}_2 = p\text{-Cl}; t = 1 \text{ h}; \text{Yield} = 7\%$;

Supplementary Fig. 45 Substrate scope of the solvent-free selective hydrogenation of different nitroaromatic compounds to unsymmetrical azoxy compounds. The yield was based on $\text{R}_1\text{-C}_6\text{H}_4\text{NO}_2$.

6. Some relevant papers shall be cited, eg. Advanced Materials, 2008471, 2021, National Science Review 10 (1), nwac100, 2023.

Response: Done.

7. The Co-O bond length in $\text{Co}_1/\text{Nb}_2\text{O}_5$ is noticeably shorter at 1.54 Å compared to 1.75 Å in CoO (Fig. 2h). Does this correspond to the simulated bond length in DFT calculations?

Response: We thank the reviewer for this comment. The peaks at 1.54 Å and 1.75 Å are associated with the scattering path of Co-O bond in $\text{Co}_1/\text{Nb}_2\text{O}_5$ and CoO measured by XAFS, respectively. This has been commonly reported in many research papers, for example, Nature Catalysis 2022, 5, 414; Nature Communications 2023, 14, 1457; Adv. Mater. 2021, 33, 2103533; Adv. Sci. 2023, 10, 2206107. In addition, this does not correspond to the simulated bond length in the DFT calculations.

8. How do the authors confirm the products? The corresponding GC-MS data or NMR spectroscopy of the products are suggested to be provided.

Response: We employed GC and GC-MS to confirm the products and we have added the corresponding spectra to the revised Supplementary Information. Please see Supplementary Figs. 19, 29-44 in the revised Supplementary Information.

REVIEWER COMMENTS

Reviewer #1 (Remarks to the Author):

The authors revised the manuscript well and answered most of my concerned questions. However, I'm still confused about the explanation of reaction mechanism.

First, they stated that H₂ tends to undergo heterolytic dissociation on Nb₂O₅(001) with a kinetic energy barrier (E_b) of 0.88 eV (ACS Catal. 2022, 12, 4806-4812). According to the authors opinion, H₂ dissociation and PhNO₂ hydrogenation occurred on Nb₂O₅ and Co1 site, respectively. So, where did H^{δ+} and H^{δ-} in the DFT calculations? For the following hydrogenation processes, these H^{δ+} and H^{δ-} species should be cleaved from Nb₂O₅. I didn't see these processes.

Second, the dissociation barrier is only 0.03 eV on the Co (111) surface (Int. J. Hydrog. Energy 2018, 43, 5576-5590). This suggested the Co (111) surface should be covered with dissociated H atoms. However, it wasn't consistent with their DFT calculations.

Third, I suggest H/D exchange experiments should be done to confirm the difference of H₂ dissociation on Co1/Nb₂O₅ and CoNP/Nb₂O₅ catalysts.

Reviewer #2 (Remarks to the Author):

The authors have addressed the concerns raised by the referee. The revised version can now be published.

Reviewer #3 (Remarks to the Author):

The authors have thoroughly addressed the comments from this reviewer and others, resulting in a substantial improvement of the manuscript. I recommend its publication in its present form.

No. NCOMMS-23-33366A

Title: Solvent-free selective hydrogenation of nitroaromatics to azoxy compounds over Co single atoms decorated on Nb₂O₅ nanomeshes

Author(s): Zhijun Li^{1,*} Xiaowen Lu,^{1,†} Cong Guo,^{2,†} Siqi Ji,¹ Hongxue Liu,¹ Chunmin Guo,¹ Xue Lu,¹ Chao Wang,³ Wensheng Yan,³ Bingyu Liu,⁴ Wei Wu,⁴ J. Hugh Horton,^{1,5} Shixuan Xin,¹ Yu Wang^{2,*}

Journal: *Nature Communications*

We would like to thank all the reviewers and editors for your time and efforts in evaluating our manuscript. We have made all of the requested changes, which are listed point-by-point in this letter (*Italic, blue*).

Response to Referee

Reviewer #1: The authors revised the manuscript well and answered most of my concerned questions. However, I'm still confused about the explanation of reaction mechanism.

First, they stated that H₂ tends to undergo heterolytic dissociation on Nb₂O₅(001) with a kinetic energy barrier (E_b) of 0.88 eV (ACS Catal. 2022, 12, 4806-4812). According to the authors opinion, H₂ dissociation and PhNO₂ hydrogenation occurred on Nb₂O₅ and Co1 site, respectively. So, where did H^{δ+} and H^{δ-} in the DFT calculations? For the following hydrogenation processes, these H^{δ+} and H^{δ-} species should be cleaved from Nb₂O₅. I didn't see these processes.

*Response: Thanks for the reviewer's insightful comment. During the revision, we calculated the kinetic energy barrier (E_b) of the dissociation of an H₂ molecule on Nb₂O₅ and Co₁-Nb₂O₅. As shown in Supplementary Fig. 55, the E_b of Nb₂O₅ is 0.91 eV, which is consistent with the previous study (0.88 eV) (ACS Catal. 2022, 12, 4806-4812), and the E_b of Co₁-Nb₂O₅ is close to that of Nb₂O₅ (Supplementary Fig. 56). This may be attributed to the fact that H₂ is physically adsorbed on Nb₂O₅ and Co₁-Nb₂O₅, of which the adsorption energy E_{H₂} is -0.17 and -0.24 eV, respectively. Please note that in both Nb₂O₅ and Co₁-Nb₂O₅, the H₂ molecule is preferentially located above the Nb site (Supplementary Fig. 57), and H₂ above the Co₁ site is also physical adsorption while being weaker (-0.12 eV). Therefore, H₂ dissociation and PhNO₂ hydrogenation may occur on the Nb₂O₅ and the Co₁ site, respectively. As H₂ is physical adsorption, we also considered H₂ dissociation on *PhNO₂ of Co₁-Nb₂O₅ via the Eley-Rideal (ER) mechanism. In this case, H₂ is physically adsorbed above the Nb site near Co₁, and PhNO₂ is chemically adsorbed on Co₁. During the H₂ dissociation, one H atom (H^{δ+}) transfers to *PhNO₂ to form *PhNO₂H, and the other one (H^{δ-}) is chemically adsorbed on the Nb site; according to our calculation, the corresponding E_b is 0.34 eV (Supplementary Fig. 60). Please note that the supplementary calculations in the last revision have shown that the hydrogenation of *PhNO₂H to *PhNO via the H^{δ-} species adsorbed on the Nb site is kinetically favorable (Fig. 5d); thus, the cleaving of H^{δ+} and H^{δ-} species from Co₁-Nb₂O₅ to intermediates is favorable.*

The above discussion has been added to the main text of the revised manuscript (Pages 11 and 12), and Supplementary Figs. 55-58, 60 have been added to the revised Supplementary Information "We also performed a kinetic analysis to gain more insights into the reactivity of Co₁/Nb₂O₅ and Co(111). It was found that the kinetic barrier (E_b) of the dissociation of an H₂ molecule on Nb₂O₅ is 0.91 eV (Supplementary Fig. 55), consistent with the previous report (0.88 eV)⁶⁴. The E_b of

$\text{Co}_1/\text{Nb}_2\text{O}_5$ (0.84 eV) is close to that of Nb_2O_5 (Supplementary Fig 56); this may be due to the fact that H_2 is physically adsorbed on Nb_2O_5 and $\text{Co}_1\text{-Nb}_2\text{O}_5$, of which the adsorption energy E_{H_2} is -0.17 and -0.24 eV, respectively. Specially, in both Nb_2O_5 and $\text{Co}_1\text{-Nb}_2\text{O}_5$, H_2 is preferentially located above the Nb site (Supplementary Fig. 57), and H_2 above the Co_1 site is also physical adsorption and weaker (-0.12 eV). Therefore, H_2 dissociation and PhNO_2 hydrogenation may occur on the Nb_2O_5 and the Co_1 site, respectively. Moreover, as expected, Co(III) exhibits stronger H_2 adsorption than $\text{Co}_1\text{-Nb}_2\text{O}_5$ and Nb_2O_5 (Supplementary Fig. 57), and its E_b is as low as 0.02 eV (Supplementary Fig. 58), which is in agreement with previous studies (0.03 eV)⁶⁵. In the presence of $^*\text{PhNO}_2$, H_2 dissociation on Co(III) is still kinetically favorable (Supplementary Fig. 59). These modeling results are consistent with our H_2 -TPD measurements and H_2 - D_2 exchange experiments. Note that the small E_b of H_2 dissociation on Co(III) should not affect the PhNO_2 adsorption and the subsequent hydrogenation steps, given its stronger adsorption of PhNO_2 than H_2 . On a different note, as H_2 is physical adsorption, we also calculated H_2 dissociation on $^*\text{PhNO}_2$ of $\text{Co}_1\text{-Nb}_2\text{O}_5$ via the Eley–Rideal (ER) mechanism and found a small E_b of 0.34 eV (Supplementary Fig. 60).”

Supplementary Fig. 55 DFT calculation of H_2 dissociation on Nb_2O_5 . The inserts are the geometric structures of the initial state (IS), transition state (TS), and final state (FS).

Supplementary Fig. 56 DFT calculation of H_2 dissociation on $\text{Co}_1\text{-Nb}_2\text{O}_5$. The inserts are the geometric structures of the initial state (IS), transition state (TS), and final state (FS).

Supplementary Fig. 57 Top and side views of an H_2 molecule adsorbed over (a) Nb_2O_5 , (b) $Co_1-Nb_2O_5$ (Nb site), (c) $Co_1-Nb_2O_5$ (Co_1 site), and (d) $Co(111)$. The numbers represent the distance between H_2 and the active site. Please note only $Co(111)$ is chemical adsorption, and the rest of cases are physical adsorption.

Supplementary Fig. 58 DFT calculation of H_2 dissociation on $Co(111)$. The inserts are the geometric structures of the initial state (IS), transition state (TS), and final state (FS).

Supplementary Fig. 59 DFT calculation of H_2 dissociation on $Co(111)$ in the presence of $*PhNO_2$. The inserts are the geometric structures of the initial state (IS), transition state (TS), and final state (FS).

Supplementary Fig. 60 DFT calculation of H_2 dissociation on $*PhNO_2$ of Co_1/Nb_2O_5 via the Eley-Rideal (ER) mechanism. The inserts are the geometric structures of the initial state (IS), transition state (TS), and final state (FS). In this case, H_2 is physically adsorbed above the Nb site near Co_1 , and $PhNO_2$ is chemically adsorbed on Co_1 ; after H_2 dissociation, one H atom ($H^{\delta+}$) transfers to $*PhNO_2$ to form $*PhNO_2H$, and the other one ($H^{\delta-}$) is chemically adsorbed on the Nb site. Please note that the hydrogenation of $*PhNO_2H$ to $*PhNO$ via the $H^{\delta-}$ species adsorbed on the Nb site is kinetically favorable (Fig. 5d); thus, the cleaving of $H^{\delta+}$ and $H^{\delta-}$ species from $Co_1-Nb_2O_5$ to intermediates is favorable.

Second, the dissociation barrier is only 0.03 eV on the Co (111) surface (Int. J. Hydrog. Energy 2018, 43, 5576-5590). This suggested the Co (111) surface should be covered with dissociated H atoms. However, it wasn't consistent with their DFT calculations.

Response: Thanks for the reviewer's comment. In principle, both the reactants, i.e., PhNO₂ and H₂, can be adsorbed on the Co(111) surface during the reaction. We found that the PhNO₂ adsorption energy on Co(111) is -2.57 eV (Fig. 5e), which is stronger than that of H₂ on Co(111) (-0.40 eV) (Supplementary Fig. 57), indicating its favorable adsorption of PhNO₂ over H₂. In the presence of *PhNO₂, H₂ dissociation on Co(111) is still kinetically favorable (Supplementary Fig. 59); its kinetic energy barrier is only 0.03 eV, which is slightly higher than that of pristine Co(111) (0.02 eV) (Supplementary Fig. 58). Please also note that the E_b of pristine Co(111) is in agreement with previous studies (0.03 eV) (Int. J. Hydrog. Energy 2018, 43, 5576-5590). Therefore, the small H₂ dissociation barrier should not affect the adsorption of PhNO₂ on Co(111) and the subsequent hydrogenation steps.

The above discussion has been added to the main text of the revised manuscript, please see Page 11 of the revised manuscript and Supplementary Figs. 57-59 in the revised Supplementary Information "Moreover, as expected, Co(111) exhibits stronger H₂ adsorption than Co₁-Nb₂O₅ and Nb₂O₅ (Supplementary Fig. 57), and its E_b is as low as 0.02 eV (Supplementary Fig. 58), which is in agreement with previous studies (0.03 eV)⁶⁵. In the presence of *PhNO₂, H₂ dissociation on Co(111) is still kinetically favorable (Supplementary Fig. 59). These modeling results are consistent with our H₂-TPD measurements and H₂-D₂ exchange experiments."

Supplementary Fig. 57 Top and side views of an H₂ molecule adsorbed over (a) Nb₂O₅, (b) Co₁/Nb₂O₅ (Nb site), (c) Co₁/Nb₂O₅ (Co₁ site), and (d) Co(111). The numbers represent the distance between H₂ and the active site. Please note only Co(111) is chemical adsorption, and the rest of cases are physical adsorption.

Supplementary Fig. 58 DFT calculation of H_2 dissociation on Co(111). The inserts are the geometric structures of the initial state (IS), transition state (TS), and final state (FS).

Supplementary Fig. 59 DFT calculation of H_2 dissociation on Co(111) in the presence of $*PhNO_2$. The inserts are the geometric structures of the initial state (IS), transition state (TS), and final state (FS).

Third, I suggest H/D exchange experiments should be done to confirm the difference of H_2 dissociation on Co_1/Nb_2O_5 and $CoNP/Nb_2O_5$ catalysts.

Response: Thanks for the reviewer's important suggestion and we have performed H_2 -TPD and H_2 - D_2 exchange experiments to show the difference of H_2 dissociation on Co_1/Nb_2O_5 and $CoNPs/Nb_2O_5$ catalysts. Please see Pages 8 and 9 in the revised manuscript and Supplementary Figs. 28 and 29 in the revised Supplementary Information "H₂ dissociation ability over catalysts plays a crucial role in the hydrogenation reactions"⁵⁷⁻⁵⁹. H_2 -temperature-programmed desorption (H_2 -TPD) measurements were initially performed and the results (Supplementary Fig. 28) show that $CoNPs/Nb_2O_5$ exhibits a higher intensity of desorption peaks over Co_1/Nb_2O_5 and Nb_2O_5 , implying the existence of a higher amount of active sites and greater H_2 adsorption capacity (Supplementary Table 1). The desorption temperature of Co_1/Nb_2O_5 is slightly smaller than that of $CoNPs/Nb_2O_5$,

but higher than that of Nb_2O_5 . Based on Kyriakou's work⁶⁰, the H_2 dissociation barrier will be lower on the active sites once the corresponding binding energy of dissociated H atoms is higher. Therefore, the higher H_2 desorption temperature of Co NPs/ Nb_2O_5 suggests that it favors the activation and dissociation of H_2 more efficiently than $\text{Co}_1/\text{Nb}_2\text{O}_5$ and Nb_2O_5 . The H_2 dissociation activity of the samples was further evaluated using an H_2 - D_2 exchange experiment (Supplementary Fig. 29). The HD formation rate follows the order Co NPs/ Nb_2O_5 > $\text{Co}_1/\text{Nb}_2\text{O}_5$ > Nb_2O_5 . Co NPs/ Nb_2O_5 achieved a higher HD formation rate than those of $\text{Co}_1/\text{Nb}_2\text{O}_5$ and Nb_2O_5 , suggesting that the addition of Co species could considerably enhance the H_2 dissociation activity and subsequently promote the hydrogenation reactions. Although Co NPs/ Nb_2O_5 exhibits excellent nitrobenzene conversion, its extremely poor azoxybenzene selectivity is observed. This implies that the difference in H_2 dissociation activity might not be the only reason affecting the overall catalytic performance of the catalyst. More discussion on the origin of the selectivity difference can be found in the kinetics simulations of key hydrogenation steps in the Mechanism investigation section." and Page 11 in the revised manuscript "These modeling results are consistent with our H_2 -TPD measurements and H_2 - D_2 exchange experiments."

Supplementary Fig. 28 H_2 -TPD profiles of Co NPs/ Nb_2O_5 , $\text{Co}_1/\text{Nb}_2\text{O}_5$, and Nb_2O_5 .

Supplementary Fig. 29 H_2 - D_2 exchange results of Co NPs/ Nb_2O_5 , $\text{Co}_1/\text{Nb}_2\text{O}_5$, and Nb_2O_5 .

Supplementary Table 1. Physical properties of samples.

Samples	Co loading ^a (wt %)	Co dispersion ^b (%)		S_{BET}^c ($m^2 \cdot g^{-1}$)	V_{pore}^d ($cm^3 \cdot g^{-1}$)	D_{pore}^e (nm)	H_2 adsorption ^f ($\mu mol/g$)
		fresh	used				
Nb_2O_5	–	–	–	49.8	0.16	14.8	38.3
Co_1/Nb_2O_5	0.42	97	93	56.7	0.19	14.5	53.4
Co NPs/Nb_2O_5	5.12	39	33	80.0	0.19	12.3	61.4

^aCobalt loading was determined by ICP-AES; ^bMetal dispersion was measured by CO chemisorption; ^c S_{BET} was determined by BET method; ^dVolume of N_2 at $P/P_0=0.99$; ^eAverage pore diameter; ^fAmount of H_2 adsorption derived from H_2 -TPD.

Reviewer #2: The authors have addressed the concerns raised by the referee. The revised version can now be published.

Response: We thank the reviewer for his/her positive feedback.

Reviewer #3: The authors have thoroughly addressed the comments from this reviewer and others, resulting in a substantial improvement of the manuscript. I recommend its publication in its present form.

Response: We appreciate the reviewer for recommending the publication of our work. Many thanks!

REVIEWERS' COMMENTS

Reviewer #1 (Remarks to the Author):

The authors addressed all the questions and it can be published in NC.